# Purkinje cell dopaminergic inputs to astrocytes regulate cerebellar-dependent behavior

Chang Li [1,4], Natalie B. Saliba[1,4], Hannah Martin[1,2], Nicole A. Losurdo[1,3], Kian Kolahdouzan[1], Riyan Siddiqui[1], Destynie Medeiros[1] & Wei Li [1] ✉

Dopamine has a significant role in motor and cognitive function. The dopaminergic pathways originating from the midbrain have received the most attention; however, the relevance of the cerebellar dopaminergic system is largely undiscovered. Here, we show that the major cerebellar astrocyte type Bergmann glial cells express D1 receptors. Dopamine can be synthesized in Purkinje cells by cytochrome P450 and released in an activity-dependent fashion. We demonstrate that activation of D1 receptors induces membrane depolarization and $Ca^{2+}$ release from the internal store. These astrocytic activities in turn modify Purkinje cell output by altering its excitatory and inhibitory synaptic input. Lastly, we show that conditional knockout of D1 receptors in Bergmann glial cells results in decreased locomotor activity and impaired social activity. These results contribute to the understanding of the molecular, cellular, and circuit mechanisms underlying dopamine function in the cerebellum, revealing a critical role for the cerebellar dopaminergic system in motor and social behavior.

Dopamine (DA) is a neuromodulator that has a profound role in motor and cognitive function[1]. A great deal of knowledge has been gained by exploring DA action in the DAergic pathways that originate from the midbrain. DA dysfunction is implicated in the neuropathology of a myriad of neurological and psychiatric diseases, including Parkinson's disease (PD), Huntington's disease (HD), schizophrenia, drug addiction, attention-deficit/hyperactivity disorder (ADHD), and autism spectrum disorder (ASD)[2–7]. In addition to the midbrain, many of these disorders are associated with an impairment in the cerebellum[8–11]. This role could be ascribed to cerebellar influence on those DAergic pathways via their anatomical connection[12]. The cerebellum has direct projections to the ventral tegmental area (VTA) and modulates its DA release, contributing to reward and social behavior[13,14]. The cerebellum also affects food consumption by modulating DA release in the striatum[15]. Nevertheless, it is also plausible that an innate DAergic system in the cerebellum is involved in the etiology of these

disorders[16]. All in all, the cerebellar DAergic system was often underappreciated, but accumulating evidence suggests it may have a significant role.

DAergic fibers were shown to be spread throughout the molecular layer (ML) in early studies using immunostaining and autoradiographic mapping in the rodent cerebellum[17,18]. In vivo autoradiography and positron emission tomography (PET) in primates demonstrated DA binding in the cerebellar dentate nucleus[19]. DA release was also detected in the rat cerebellum using microdialysis[20]. Quantitative analysis by high-performance liquid chromatography (HPLC) demonstrated cerebellar DA[21], consistent with a more recent finding that revealed cerebellar DA using proton nuclear magnetic resonance ($^1$H-NMR)[22]. Physiological experiments further showed that DA signaling is responsible for $Ca^{2+}$-mediated depolarization-induced slow currents (DISCs) in Purkinje cells (PCs)[23]. DA receptor subunits D1R-D5R have also been observed in the various lobules of the cerebellar cortex[16,24].

[1]Department of Neurobiology, University of Alabama at Birmingham, Birmingham, AL, USA. [2]Present address: Department of Psychiatry and Psychotherapy, University Medical Center Göttingen, Göttingen, Germany. [3]Present address: Neuroscience Program, The University of Utah, Salt Lake City, UT, USA. [4]These authors contributed equally: Chang Li, Natalie B Saliba. ✉e-mail: weili7@uab.edu

These findings indicate that an independent DAergic system exists in the cerebellum. However, the exact source of DA and the precise distribution of DA receptors are undefined.

The cerebellum is a unique brain structure, where PCs integrate excitatory synaptic inputs from climbing fibers (CFs) and parallel fibers (PFs) and send solely inhibitory output[25]. Intercalated among PC soma and dendrites are molecular layer interneurons (MLIs) and Bergmann glial cells (BGs). MLIs provide a major synaptic inhibition and BGs have a critical role in fine control of cerebellar neurotransmission and network activity. BGs are the primary astrocyte type and outnumber PCs by approximately eightfold[26]. Each BG has up to five polarized processes extending through the entire ML. These radial BG fibers sprout small, convoluted side branches that generate lamellate appendages, forming neuron-glial interaction sites. By ensheathing excitatory and inhibitory synapses on PCs, BGs are crucial for neuroprotection, synaptic stability, and synaptic plasticity[27–29], and thus contribute to neuropathological mechanisms in many diseases[30–32]. DA receptor subtypes are known to be distributed in neuronal types, while our finding reveals that D1Rs are also highly expressed in BGs. Our experiments were designed to assess DAergic modulation of BGs, its effect on PCs, and its role in motor and non-motor behaviors.

## Results

### DAergic system in the cerebellum

*Drd1a*-tdTomato and *Drd2*-EGFP transgenic mice have been widely utilized to characterize the role of D1Rs and D2Rs in the basal ganglia[33,34]. Using these mice, we found that the expression level of D2R reporter GFP was minimal in the cerebellum; however, the D1R reporter tdTomato was highly enriched (Fig. 1a, Supplementary Fig. 1a). The D1R expression was observed in cells with multiple radial branches, suggestive of cerebellar BGs (Fig. 1a, right). Indeed, immunostained cerebellar sections demonstrated colocalization of tdTomato with the astrocyte marker glial fibrillary acidic protein (GFAP) or S100β (Fig. 1b and Supplementary Fig. 1b), but not with the PC marker calbindin D28K (CB) (Supplementary Fig. 1c), confirming the expression of D1Rs in BGs. As the astrocyte markers only label the soma and

the main processes, to fully demonstrate the distribution of D1Rs in BGs, we filled BGs with biocytin followed by tdTomato immunostaining (Fig. 1c). We found that the fine processes of BGs were also enriched with D1Rs. We further performed immunostaining with an antibody selective for D1Rs, which had been demonstrated in the striatum by their overlap with the D1R reporter but not the D2R reporter (Supplementary Fig. 1d). Co-labeling with tdTomato showed that D1Rs were expressed in BG soma and processes (Fig. 1d). Similarly, immunolabelled D1Rs were observed in BGs filled with biocytin (Fig. 1e). Such distribution was also supported by single-molecule fluorescence in situ hybridization (smFISH), showing *Drd1a* mRNA expression at the border of the Purkinje cell layer (PCL) and molecular layer (ML) (Fig. 1f). We further revealed that the DA transporter (DAT) was also expressed in BGs and PCs (Fig. 1g), by using an antibody selective for DAT1 (Supplementary Fig. 1e). In the cerebellar granule cell layer (GCL), tdTomato was not found in NeuN-positive GCs or GFAP-positive astrocytes (Supplementary Fig. 2a). In the deep cerebellar nuclei, tdTomato was identified in some cells but not in GFAP-positive astrocytes (Supplementary Fig. 2b). Other than the cerebellum, co-immunostaining of tdTomato with GFAP or S100β demonstrated that D1R-expressing cells were generally not astrocytes, as shown in several major brain regions including the cerebral cortex, hippocampus, and striatum (Supplementary Fig. 2c-h).

### DA source in the cerebellum

We next sought to identify the source of DA within the cerebellum. DA synthesis is generally thought to involve the rate-limiting enzyme tyrosine hydroxylase (TH)[35]. Using TH as the marker, we found that DAergic neurons were largely localized in the substantia nigra pars compacta (SNc), VTA, as well as the locus coeruleus (LC) (Supplementary Fig. 3a, b), which is consistent with the reported DA sources in the brain[36]. To determine if their axonal terminals enter the cerebellum, we injected AAV-DIO-YFP vectors into the SNc/VTA and LC of *Slc6a3*-Cre mice to express YFP in DAergic neurons (Supplementary Fig. 3c–f). The axonal bundles emanating from the SNc/VTA were clearly seen entering the cerebellum through the superior cerebellar

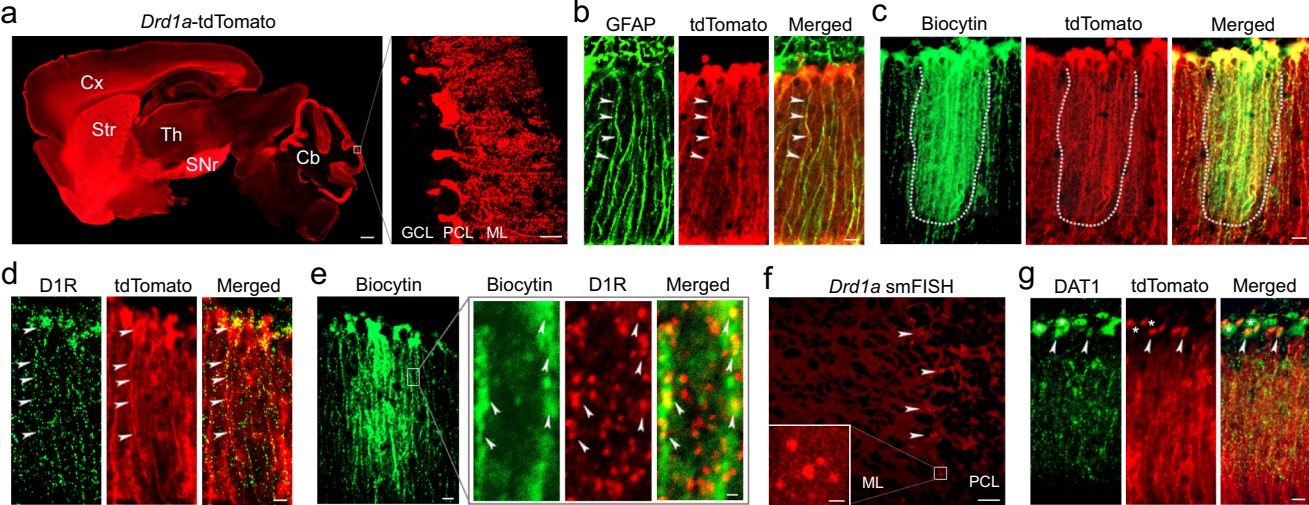

**Fig. 1 | Expression of D1Rs in BGs of the cerebellum. a** A sagittal brain section from a *Drd1a*-tdTomato mouse. Five biological replicates were performed. Cx, cerebral cortex; Str, striatum; Th, thalamus; SNr, substantia nigra pars reticulata; Cb, cerebellum; GCL, granule cell layer; PCL, Purkinje cell layer; ML, molecular layer. Scale bar, 500 μm (left) and 50 μm (right). **b** Dual immunostaining of GFAP and tdTomato. Arrowheads indicate their colocalization. Five biological replicates were performed. Scale bar, 20 μm. **c** BG filled with biocytin followed by tdTomato immunostaining. Dotted outlines indicate their colocalization along the entire processes of BGs. Four biological replicates were performed. Scale bar, 20 μm.

**d** D1R expression in *Drd1a*-tdTomato BGs (arrowheads). Four biological replicates were performed. Scale bar, 20 μm. **e** BG filled with biocytin followed by D1R immunostaining. Arrowheads indicate their colocalization. Four biological replicates were performed. Scale bar, 20 μm (left) and 2 μm (right, enlarged images). **f** *Drd1a* smFISH shows expression of *Drd1a* mRNAs at the border of the PCL and ML. Scale bar, 50 μm (inset, 10 μm). Three biological replicates were performed. **g** DAT1 expression in *Drd1a*-tdTomato sections. Arrowheads indicate BG soma; asterisks indicate PC soma. Four biological replicates were performed. Scale bar, 20 μm.

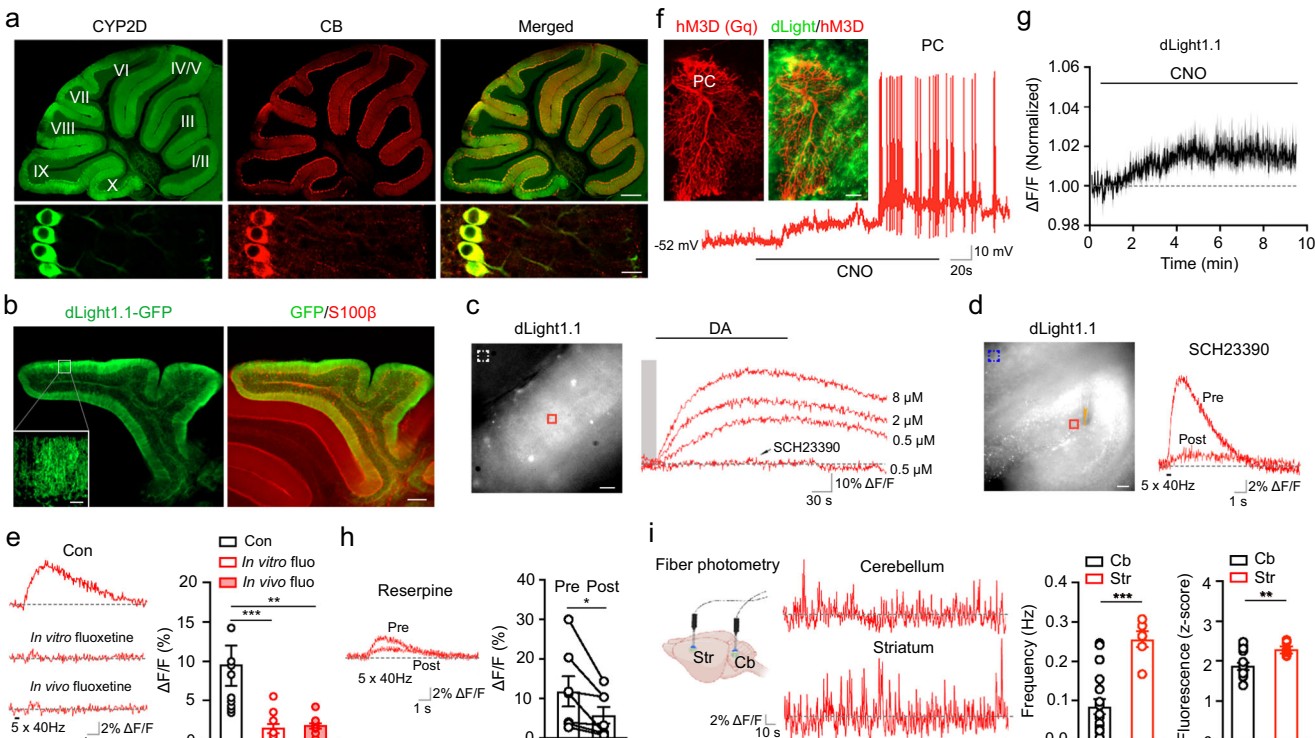

**Fig. 2 | DA source in the cerebellum. a** Dual immunostaining of CYP2D and CB. Six biological replicates were performed. Scale bar, 500 μm (upper) and 20 μm (lower). **b** Dual immunostaining of GFP and S100β. CAG-dLight1.1-GFP was expressed in the cerebellum. Three biological replicates were performed. Scale bar, 400 μm (inset, 50 μm). **c** dLight1.1 signals induced by DA at different concentrations (n = 3 slices/1 mouse) and blocked by SCH23390. The solid red box shows ROIs where dLight1.1 signals were measured; the dashed white box indicates background signals. **d** dLight1.1 signals evoked by electrical stimulus were recorded before and 15 min after SCH23390 application (n = 4 slices/2 mice). Scale bar, 50 μm. The solid red box shows ROIs; the dashed blue box indicates background signals. The orange arrow points to the placement of the stimulus electrode. **e** dLight1.1 signals evoked in slices perfused with normal aCSF (n = 10 slices/3 mice), in slices pretreated with fluoxetine (n = 10 slices/3 mice), or in slices from mice administered with fluoxetine (n = 11 slices/3 mice). Data are presented as mean ± SEM. Kruskal–Wallis test with

Dunn's multiple comparisons test (p = 0.0003, Con vs. In vitro fluo; p = 0.003, Con vs. In vivo fluo). **f** Dual immunostaining of mCherry and GFP. Example trace of a whole-cell recording was acquired from a hM3D(Gq)-expressing PC. Scale bar, 20 μm. **g** dLight1.1 signals induced by CNO application (n = 17 slices/5 mice). The solid line indicates the mean, and the shaded area shows the SEM. Unpaired Student's t test (p = 0.0357, baseline vs. 6 min after CNO). **h** dLight1.1 signals were recorded before and 10 min after reserpine application (n = 7 slices/3 mice). Data are presented as mean ± SEM. Two-sided Wilcoxon matched pairs signed rank test (p = 0.0156, Pre vs. Post). **i** In vivo dLight1.1 signals recorded in the cerebellum (n = 22 recordings/5 mice) and striatum (n = 6 recordings/3 mice). Data are presented as mean ± SEM. Unpaired two-sided Student's t test (p < 0.0001, Hz Cb vs. Str; p = 0.0064, Fluorescence Cb vs. Str). The cartoon was created with BioRender.com.

peduncle (SCP) but did not appear to extend into the cerebellar cortex. (Supplementary Fig. 3c–e). YFP-expressing axons from the LC also did not project into the cerebellum (Supplementary Fig. 3f). To further explore these potential pathways, we injected an anterograde fluorescent tracer Dextran-Alexa Fluor 488 into the SNc/VTA and allowed 4 days for its transport (Supplementary Fig. 3g). As shown by YFP expression, the axonal afferents were identified in the SCP but not in the cerebellar cortex (Supplementary Fig. 3h–j). We also injected Retrobeads into crus 1 and 2 of the cerebellum; however, we did not observe apparent retrogradely labeled DAergic neurons in the SNc/VTA (Supplementary Fig. 3k, l). These results indicate that the SNc, VTA, or LC may not be the major source of DA supplied to the cerebellum.

We then explored a member of the cytochrome P450 (CYP) superfamily CYP2Ds that can metabolize tyramine into DA[37–40], and are found in the cerebellum[41]. Immunostaining using antibodies against N or C terminal CYP2D6 and CB demonstrated that CYP2D was intensely expressed in CB-positive PC dendrites and soma (Fig. 2a and Supplementary Fig. 4a), suggesting that cerebellar DA may be synthesized in this alternative pathway. To detect the DA release following its synthesis, we injected AAV-CAG-dLight1.1 into the cerebellar cortex to express DA biosensor dLight1.1 (Fig. 2b)[42]. dLight1.1 was highly expressed in S100β-positive BGs but to a lesser

degree in CB-positive PCs (Supplementary Fig. 4b). To confirm its sensitivity, we applied exogenous DA to cerebellar slices at different concentrations (0.5, 2, and 8 μM). dLight1.1 signals were robustly evoked in a dose-dependent manner, which was blocked by pretreatment with the D1 antagonist SCH23390 (Fig. 2c). We further assessed activity-dependent DA release by delivering electrical stimuli to dLight1.1-expressing cerebellar slices. Strong dLight1.1 signals were evoked and then abolished following SCH23390 treatment (Fig. 2d). To pharmacologically verify the role of CYP2D in DA synthesis, we performed in vitro and in vivo treatments with a potent CYP2D inhibitor fluoxetine[43,44]. We expressed dLight1.1 in the cerebellum and later prepared slices for treatment with fluoxetine for 2 h. In a different set of experiments, during dLight1.1 expression, we also administered fluoxetine to mice for 2 wks prior to slice preparation. Intriguingly, in both groups evoked dLight1.1 signals were largely reduced compared with the control (Fig. 2e). A similar effect was seen in slices following in vitro treatments of another CYP2D inhibitor quinidine (Supplementary Fig. 4c)[45,46]. To exclude the possibility that fluoxetine treatment degraded dLight1.1 itself, we delivered stimuli followed by DA perfusion to the same slice. In the groups that had undergone in vitro and in vivo treatments, stimuli did not trigger dLight1.1 signals, but the following exogenous DA still induced signals (Supplementary Fig. 4d). These data suggest that DA

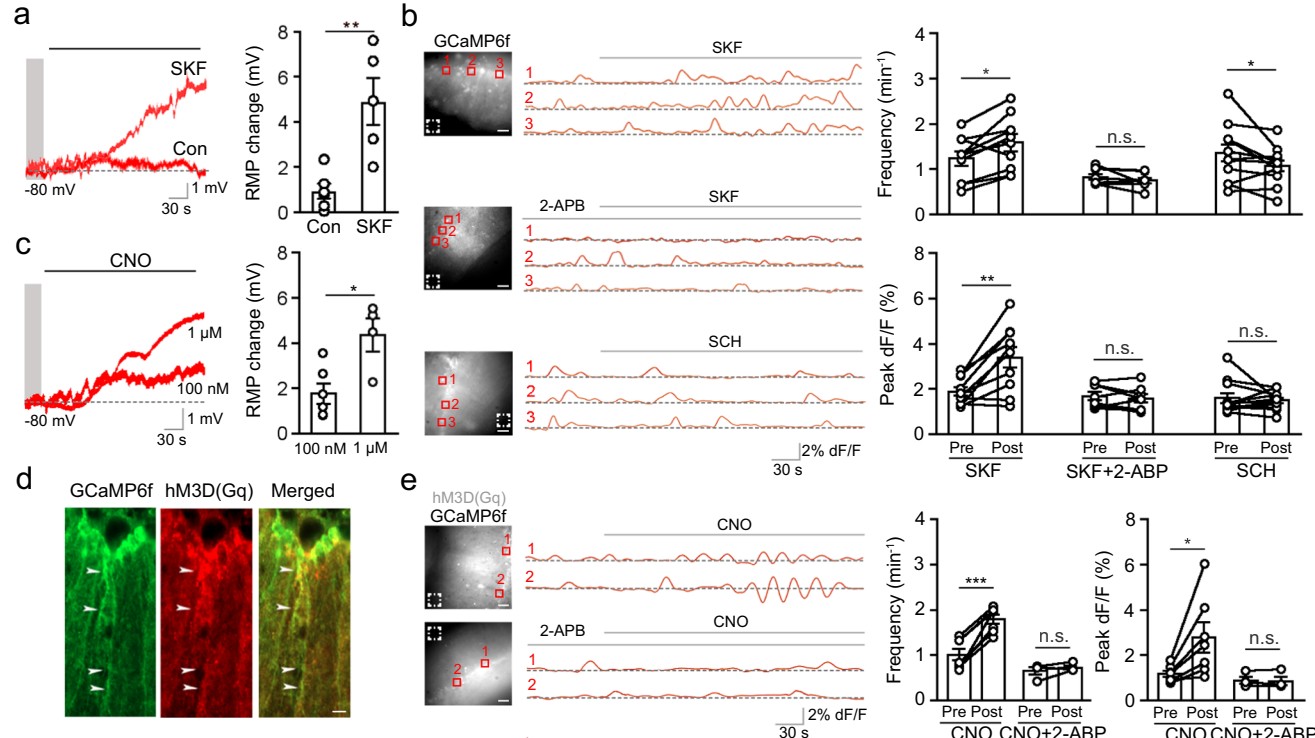

**Fig. 3 | D1R activation results in BG membrane depolarization and elevated Ca2+ signals. a** Membrane depolarization induced by SKF83822 in BGs ($n = 6$ slices/2 mice, Con; $n = 5$ slices/3 mice, SKF). Data are presented as mean ± SEM. Mann–Whitney two-sided test ($p = 0.0144$). **b** Spontaneous $Ca^{2+}$ signals recorded in GCaMP6f-expressing BGs treated with SKF83822 ($n = 10$ slices/4 mice), or SKF83822 in the presence of 2-APB ($n = 8$ slices/4 mice), or SCH23390 ($n = 12$ slices/4 mice). AAV5-gfaABC1D-GCaMP6f was injected into the cerebellum for expression of GCaMP6f in BGs. The solid red box shows ROIs where dLight1.1 signals were measured; the dashed box indicates background signals that were subtracted. Scale bar, 50 μm. Data are presented as mean ± SEM. Paired two-sided Student's $t$ test ($p = 0.0107$, Frequency SKF; $p = 0.375$, Frequency SKF + 2-ABP; $p = 0.0391$, Frequency SCH; $p = 0.0022$, Peak dF/F SKF; $p = 0.4615$, Peak dF/F

SKF + 2-ABP; $p = 0.6290$, Peak dF/F SCH). **c** Membrane depolarization induced by CNO (100 nM and 1 μM) in hM3D(Gq)-expressing BGs ($n = 6$ slices/3 mice, CNO 100 nM; $n = 4$ slices/2 mice, CNO 1 μM). Paired two-sided Student's $t$ test ($p = 0.0128$, 100 nM vs. 1 μM). **d** Colocalization (arrowheads) of gfaABC1D-GCaMP6f-GFP and GFAP-hM3D(Gq)-mCherry immunostaining. Four biological replicates were performed. Scale bar, 20 μm. **e** Spontaneous $Ca^{2+}$ signals recorded in hM3D(Gq)-expressing BGs treated with CNO ($n = 7$ slices/3 mice), or CNO in the presence of 2-APB ($n = 4$ slices/2 mice). Scale bar, 50 μm. Data are presented as mean ± SEM. Paired two-sided Student's $t$ test ($p = 0.0001$, Frequency CNO; $p = 0.3934$, Frequency CNO + 2-ABP; $p = 0.0251$, Peak dF/F CNO; $p = 0.4114$, Peak dF/F CNO + 2-ABP). n.s., not significant.

in the cerebellum is produced in PCs through a CYP2D-dependent non-canonical pathway and released in an activity-dependent manner.

Field electrical stimulation in slices inevitably evoked the activity of a population of heterogeneous cells. To examine the immediate role of PC activity in DA release, we first attempted to record dLight1.1 signals during PC firing triggered by injections of sequential positive currents into a cell (Supplementary Fig. 4e). As action potentials from a single PC might be insufficient to induce detectable DA release, no dLight1.1 signals were seen in slices. We then employed chemogenetic stimulation to depolarize a population of PCs by expressing an excitatory designer receptor exclusively activated by a designer drug (DREADD), hM3D(Gq), in the cerebellum of *Pcp2*-Cre mice. PCs expressing hM3D(Gq) were located in close proximity to dLight1.1-expressing BGs; as expected, bath-application of the DREADD activator clozapine *N*-oxide (CNO) led to PC depolarization and firing (Fig. 2f). Examination of dLight1.1 signals revealed a significant elevation following CNO perfusion (Fig. 2g), suggesting that DA is released specifically from PCs. To further examine if DA release is related to the action of vesicular monoamine transporter (VMAT)−2, we applied the VMAT-2 inhibitor reserpine to dLight1.1-expressing slices[47]. dLight1.1 signals were significantly decreased (Fig. 2h), indicating the important role of VMAT-2 in DA release.

Lastly, we compared DA release in the cerebellum with the striatum using in vivo fiber photometry. Examination of dLight1.1 signals in

freely moving mice showed that the spontaneous signals in the cerebellum were less frequent and smaller in fluorescence z-score than those in the striatum (Fig. 2i). Altogether, these data show tonic DA release in these two brain regions is differentially regulated, which is consistent with our results showing that DA in the cerebellum is mainly originated from PCs but not from the SNc, VTA, or LC.

## DA-mediated membrane depolarization and $Ca^{2+}$ signaling in BGs

DA is known to promote cell depolarization and excitability in basal ganglia[48]. To study the effect of D1R activation on membrane potential, we performed current clamp in BGs at their resting membrane potential. Bath-application of the D1 agonist SKF83822 resulted in the depolarization of BG membrane potential (Fig. 3a). To explore the possible contribution of BG depolarization to $Ca^{2+}$ transients, we infused BGs with the organic $Ca^{2+}$ indicator Fluo 5F and then monitored $Ca^{2+}$ signals in response to membrane depolarization induced by short sequential current injections (Supplementary Fig. 5a). Compared with the control, membrane depolarization significantly increased the frequency and peak amplitude of $Ca^{2+}$ transients in BGs. We subsequently attempted to test if membrane hyperpolarization thwarts SKF83822-induced $Ca^{2+}$ signals. Interestingly, the hyperpolarization itself also increased $Ca^{2+}$ activity (Supplementary Fig. 5a)[49,50].

We further assessed the direct effect of D1R activation on $Ca^{2+}$ signals by using the genetically encoded $Ca^{2+}$ indicator gfaABC1D-

GCaMP6f. It was specifically expressed in BGs, as indicated by its colocalization with S100β in immunostaining (Supplementary Fig. 5b); it reliably reflected intense Ca$^{2+}$ dynamics in BGs (Supplementary Fig. 5c). We found that SKF83822 treatment enhanced the frequency and amplitude of spontaneous Ca$^{2+}$ signals (Fig. 3b and Supplementary Movie 1). We next tested if Ca$^{2+}$ is released from the internal store. Pretreatment with the IP3R antagonist 2-APB prevented the effect (Supplementary Movie 2), suggesting Ca$^{2+}$ rise is at least partially due to intracellular Ca$^{2+}$ mobilization. Furthermore, inhibition of D1Rs by SCH23390 attenuated the Ca$^{2+}$ frequency increase and slightly decreased the amplitude. To test if the activity of D2Rs has any effect on Ca$^{2+}$ rises in BGs, we applied the D2R agonist quinpirole to GCaMP6f-expressing slices. Imaging of Ca$^{2+}$ signals in BGs did not reveal any alterations in frequency or amplitude (Supplementary Fig. 5d), which agrees with the expression of D2R in PCs but not BGs[24]. These data suggest that D1Rs activated by DA have a significant role in eliciting Ca$^{2+}$ rises in BGs.

To test if these results can be replicated by chemogenetically depolarizing BGs, we expressed GFAP-hM3D(Gq) in BGs (Supplementary Fig. 5e). CNO treatment led to depolarization of membrane potential in BGs in a dose-dependent fashion (Fig. 3c). We then co-expressed gfaABC1D-GCaMP6f and GFAP-hM3D(Gq) in BGs (Fig. 3d). Imaging of GCaMP6f in these slices showed that CNO treatment resulted in a robust Ca$^{2+}$ oscillation in BGs (Fig. 3e). Consistent with the role of IP3 signaling in hM3D(Gq)-induced Ca$^{2+}$ fluctuation[51], 2-APB pretreatment eliminated the oscillation. These data indicate that the D1R activation is involved in membrane depolarization and IP3R-mediated Ca$^{2+}$ signaling in BGs.

## DAergic modulation of AMPARs in BGs

In addition to the fast action, D1Rs may have a role in AMPAR membrane insertion, considering previous reports on this regulation in neurons[52–54]. Therefore, we first characterized the cellular distribution of the major AMPAR subunits in the cerebellum. Dual immunostaining of AMPAR subunits and GFAP or CB showed that GluA1 and GluA4 subunits were localized to BGs, whereas GluA2 was present in PCs (Fig. 4a and Supplementary Fig. 6a), which is consistent with previous reports[28,55,56]. Immunostaining of these subunits in *Drd1a*-tdTomato mice also uncovered similar results (Supplementary Fig. 6b). We further loaded BGs with biocytin during whole-cell recordings, followed by immunostaining for GluA1, GluA2, and GluA4, and performed Airyscan confocal microscopy (Fig. 4b). Examination of these subunits on biocytin-filled processes in *x-y*, *y-z*, and *x-z* projections verified the localization of GluA1 and GluA4 in BGs and GluA2 in PCs. As AMPARs lacking GluA2 subunits are inward rectifying and permeable to Ca$^{2+}$[57], the immunostaining result suggests that AMPARs in BGs are mainly GluA2-lacking Ca$^{2+}$-permeable (CP)-AMPARs. The current-voltage (*I-V*) relationship of membrane currents evoked in BGs by direct AMPA application, indeed, exhibited the typical inward rectification (Fig. 4c). In BGs filled with the Ca$^{2+}$ indicator Oregon Green 488 BAPTA-1 (OGB-1), AMPA-induced currents in BGs were also accompanied by intracellular Ca$^{2+}$ signals (Supplementary Fig. 6c). AMPA currents and their associated Ca$^{2+}$ signals were sensitive to a selective CP-AMPAR blocker NASPM, and completely blocked by the AMPAR antagonist CNQX. These electrophysiological and Ca$^{2+}$ imaging data demonstrate that cerebellar BGs express GluA2-lacking CP-AMPARs.

We then determined whether pharmacological activation of D1Rs modulates AMPAR subunits in BGs. We treated cerebellar slices with SKF83822 for 1 h and prepared homogenates for Western immuno-blotting. SKF83822 treatment did not alter the total protein levels of GluA1 and GluA4 subunits (Fig. 4d). However, surface biotinylation assay on GluA1 and GluA4 in slices showed that SKF83822 increased surface levels of GluA1 and to a lesser degree GluA4 (Fig. 4e). This effect on GluA1 in BGs was blocked by the protein kinase A (PKA)

inhibitor Rp-cAMPs, consistent with the role of PKA in D1R-mediated GluA1 trafficking in neurons[58]. It is known that GluA1 Ser831 is phosphorylated by protein kinase C (PKC) and Ca$^{2+}$/calmodulin-dependent kinase II (CaMKII), whereas Ser845 is phosphorylated by PKA[59]. Congruent with the involvement of PKA in D1R activation, SKF83822 treatment resulted in phosphorylation of GluA1 at Ser845 but not at Ser831 (Fig. 4f). To further investigate the role of D1Rs in regulating GluA1 and GluA4 subunits, we generated *Drd1* conditional knockout (cKO) in BGs by crossing *Slc1a3*-Cre mice to *Drd1tm2.1* mice followed by administration of tamoxifen to their offspring[28,60]. Immunostaining in cerebellar sections from these mice showed the lack of D1R immunoreactivity in BGs (Supplementary Fig. 6d). Surface biotinylation assay on AMPAR subunits in the cerebellum of these mice showed that surface levels of GluA1 and GluA4 but not GluA2 were downregulated compared with controls (Fig. 4g). These data demonstrate that D1Rs have a critical role in surface insertion and homeostasis of AMPAR subunits in BGs.

## BG modulation of synaptic input to PCs through D1R activation

BGs can profoundly impact the activity of neighboring PCs by actively releasing neurotransmitters or passively altering their reuptake[26]. To study the modulation of PC activity by D1R activation, we delivered alternating stimuli to electrophysiologically-identified PFs and CFs and recorded postsynaptic currents (PSCs) in PCs during SKF83822 perfusion (Supplementary Fig. 7a, b). In particular, to examine CF input to PCs, we ensured that the afferent path was stimulated and the PSCs were not unclamped (Supplementary Fig. 7c, d). Surprisingly, without blockade of GABAergic activity, SKF83822 treatment resulted in a large reduction in the amplitude of PSCs at both PF-PC and CF-PC synapses (Supplementary Fig. 7e). It should be noted that in a few cases especially in CF-PC synapses, there existed a transient increase in PSC amplitude lasting about 1–2 min (Supplementary Fig. 7f).

We first investigated if MLIs contribute to the reduction of PSC amplitude because MLIs provide dense inhibitory innervations on PCs. Indeed, dual immunostaining of CB and parvalbumin (PV) demonstrated that PCs were innervated by a large portion of puncta derived from PV-positive MLIs (Supplementary Fig. 7g). In biocytin-filled MLIs, numerous axonal terminals were opposed to the soma and dendrites of PCs (Supplementary Fig. 7h). Considering the space clamp from distal dendrites in our recordings, the currents generated by MLI activity could thwart inward currents, leading to a decrease in PSC amplitude. Indeed, when PCs were clamped to a more depolarized state, GABA$_A$R-mediated outward currents evoked at −40 mV were evident (Supplementary Fig. 7i). Notably, when PCs were held at a similar depolarizing potential, SKF83822 increased GABA$_A$R-mediated outward currents (Supplementary Fig. 7j), suggesting that an increase in MLI activity is one factor contributing to the reduced PC excitation.

To further dissect cellular mechanisms, we recorded from PCs spontaneous inhibitory PSCs (sIPSCs) in the presence of the AMPAR antagonist NBQX, the NMDAR antagonist D-AP5, and high intracellular Cl$^-$ (140 mM) and recorded spontaneous excitatory PSCs (sEPSCs) in the presence of picrotoxin and low intracellular Cl$^-$ (17.5 mM) (Fig. 5a and Supplementary Fig. 7k). SKF83822 was shown to increase the frequency of sIPSCs although it had no effect on their amplitude. By also examining sEPSCs during SKF83822 treatment, we found that their frequency underwent a marked reduction preceded by a transient enhancement while their amplitude remained unaltered. Based on this evidence, we infer that the increase in sIPSC frequency and the decrease in sEPSC could explain the reduction of evoked PSCs during SKF83822 application. To provide chemogenetic evidence for D1R-mediated modulation of PCs, we applied CNO to hM3D(Gq)-expressing BGs and recorded PSCs following PF stimuli. CNO that exerted similarly to SKF83822 on BG Ca$^{2+}$ signals

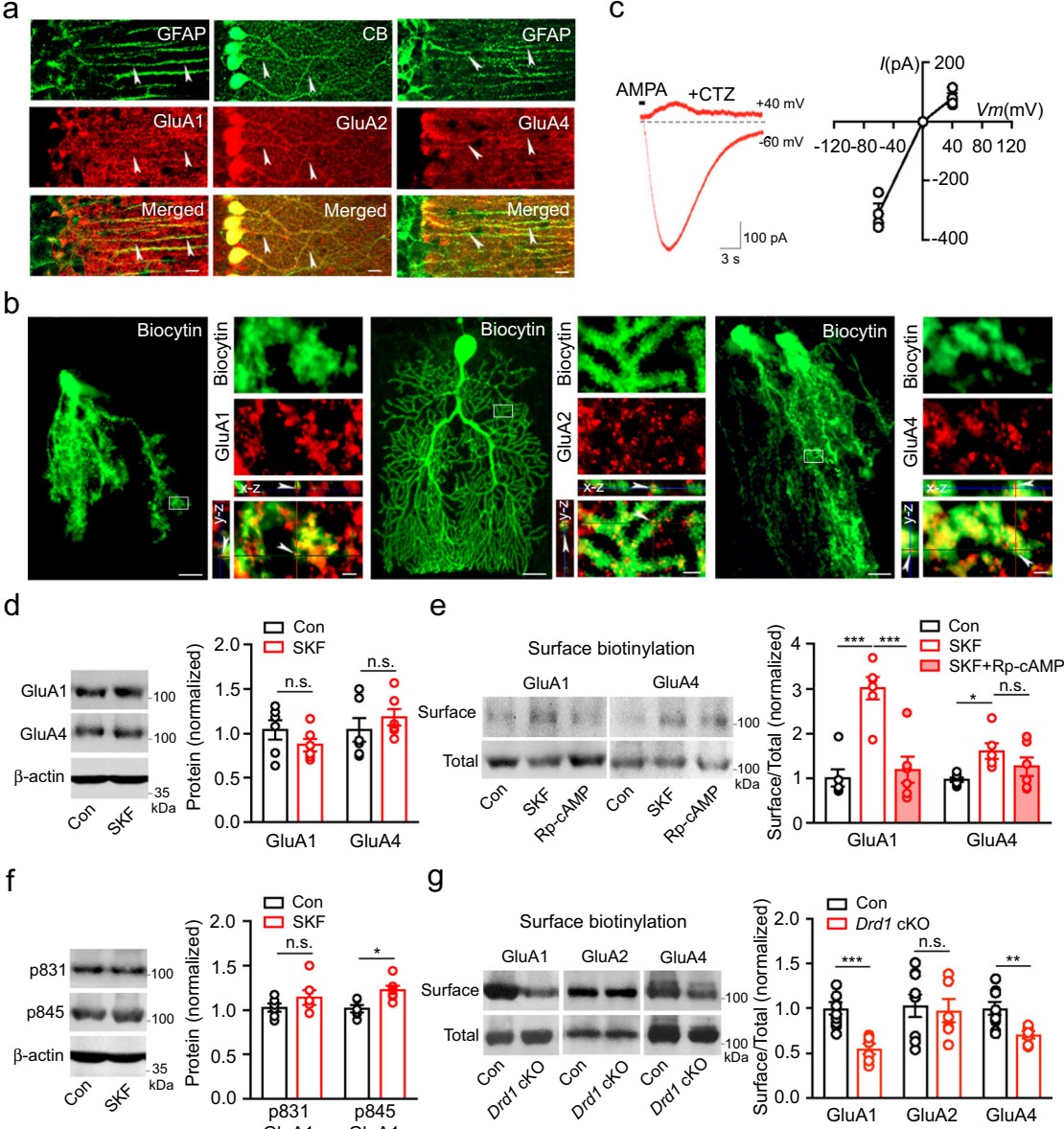

**Fig. 4 | D1R-mediated AMPAR distribution in BGs. a** Dual immunostaining of GFAP and GluA1 (left), CB and GluA2 (middle), and GFAP and GluA4 (right). Arrowheads indicate colocalization. Six biological replicates were performed. Scale bar, 20 μm. **b** BG or PC filled with biocytin followed by GluA1 (left), GluA2 (middle), and GluA4 (right) immunostaining. Scale bar, 20 μm (lower magnification) and 2 μm (higher magnification). Three biological replicates were performed for each. **c** Inward rectification of membrane currents shown in BGs (−60 and +40 mV; $n = 3$ slices/2 mice. **d** Total protein levels of GluA1 and GluA4 in the cerebellar slices treated with SKF83822 ($n = 6$, each group). Data are presented as mean ± SEM. Unpaired two-sided Student's $t$ test ($p = 0.2269$, GluA1 Con vs. SKF; $p = 0.4057$, GluA4 Con vs. SKF). **e** Levels of surface GluA1 and GluA4 proteins. Cerebellar slices were treated with vehicle (DMSO), SKF83822, or SKF83822 + Rp-cAMPs for 1 h and then subjected to GluA1 and GluA4 surface biotinylation assay ($n = 6$, each group). Data are presented as mean ± SEM. One-way ANOVA, Bonferroni's multiple comparison test and Kruskal–Wallis test with Dunn's multiple comparison test ($p = 0.0001$, GluA1 Con vs. SKF; $p = 0.0002$, GluA1 SKF vs. SKF + Rp-cAMP; $p = 0.0449$, GluA4 Con vs. SKF; $p = 0.4329$, GluA4 SKF vs. SKF + Rp-cAMP). **f** Levels of p831 GluA1 and p845 GluA1 in slices treated with SKF83822 ($n = 6$, each group). Data are presented as mean ± SEM. Unpaired two-sided Student's $t$ test ($p = 0.2667$, p831 GluA1 Con vs. SKF; $p = 0.0214$, p845 GluA1 Con vs. SKF). **g** Levels of surface GluA1, GluA2, and GluA4 in the cerebellum of control ($n = 9$) and *Drd1* cKO ($n = 6$) mice. Data are presented as mean ± SEM. Unpaired two-sided Student's $t$ test ($p = 0.0004$, GluA1 Con vs. *Drd1* cKO; $p = 0.7705$, GluA2 Con vs. *Drd1* cKO; $p = 0.0081$, GluA4 Con vs. *Drd1* cKO). n.s., not significant.

(Fig. 3c-e), also reduced the amplitude of evoked PSCs (Supplementary Fig. 7l).

Lastly, to examine the effect of BG modulation on PC output, we evaluated spontaneous PC spiking by using cell-attached recordings on PCs (Fig. 5b). We found that SKF83822 significantly reduced the frequency of spikes in PCs. In addition, the firing regularity was also decreased, as indicated by an increase in the coefficient of variation 2 (CV2) (Fig. 5c). Collectively, these data suggest that modulation of BGs by D1R activation has an overall inhibitory effect on PC activity.

## Decreased locomotor activity but intact eyeblink conditioning in *Drd1* cKO mice

To test if the DAergic system is implicated in behaviors, we conducted a battery of behavioral tests in *Drd1* cKO mice. First, we performed an open field test in *Drd1* cKO mice to evaluate locomotor and exploratory activity. Compared with control mice, *Drd1* cKO mice exhibited a lower level of locomotor activity as evaluated by the total distance traveled in the arena (Fig. 6a). Further analyses showed no difference in the percent time spent and the percent distance traveled in the central

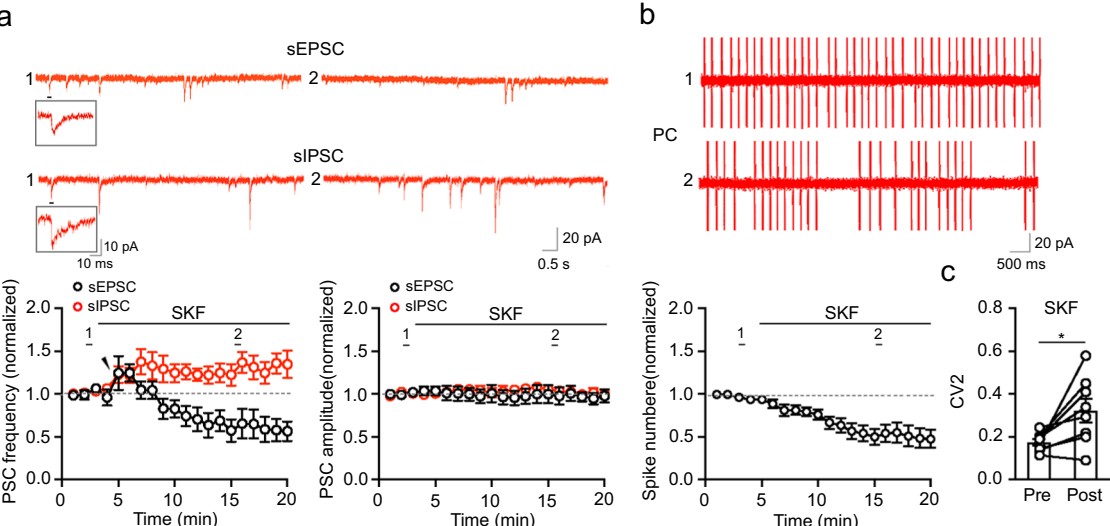

**Fig. 5 | Modulation of PC activity by D1R activation in BGs. a** sEPSCs recorded in PCs (at −60 mV) in the presence of picrotoxin and intracellular Cs-gluconate (120 mM) (*n* = 6 slices/3 mice) and sIPSCs recorded in PCs (at −60 mV) in the presence of NBQX, D-AP5, and intracellular CS-Cl (140 mM) (*n* = 6 slices/3 mice). Insets show a single sEPSC (upper) and sIPSC (lower). Data are presented as mean ± SEM. Mann−Whitney two-sided test (*p* = 0.0087, sEPSC frequency; *p* = 0.0260 sIPSC frequency; *p* = 0.8182, sEPSC amplitude; *p* = 0.8182 sIPSC amplitude; baseline vs. 15 min after SKF83822 application). Example traces before (1) and during the treatment (2) are shown. **b** Spontaneous spiking in PCs of slices treated with

SKF83822 (*n* = 10 slices/5 mice). Recordings were performed under cell-attached mode. The average spike rate among all recordings before and after SKF83822 treatments was approximately 7.9 Hz and 5.3 Hz. Data are presented as mean ± SEM. Mann−Whitney two-sided test (*p* = 0.0185, baseline vs. 10 min after SKF83822 treatment). **c** CV2 of spontaneous spiking in PCs before and during SKF83822 treatment (*n* = 8 slices/4 mice). Data are presented as mean ± SEM. Paired two-sided Student's *t* test (*p* = 0.0201 Pre vs. Post). Example traces before (1) and during the treatment (2) are shown.

area (Fig. 6a1–a3), suggesting normal anxiety levels in *Drd1* cKO mice. To test if D1R activation has an opposite effect on motility, we performed the test in mice that had received bilateral intracranial microinfusion of SKF83822 in the cerebellum for 5 days. SKF83822 microinfusion significantly enhanced the distance traveled, compared with the distance traveled prior to the treatment (Fig. 6b). Furthermore, we used the rotarod and the ladder rung walking tests to evaluate motor coordination in *Drd1* cKO mice. The latency to fall from the rotarod was significantly shorter in these mice (Fig. 6c). There was also an increasing trend for the number of foot slip errors in the ladder rung walking test (Supplementary Fig. 8a). Automated gait analysis did not reveal a significant difference in the average stride length of right forepaw (RF), right hindpaw (RH), left forepaw (LF), and left hindpaw (LH) (Supplementary Fig. 8b). We further performed grip strength test to evaluate the neuromuscular function in *Drd1* cKO. Front grip strength was significantly decreased compared with control mice (Fig. 6d). In addition, we used the marble burying test to examine repetitive digging behaviors and the nestlet building to evaluate general well-being (Supplementary Fig. 8c, d). No differences between control and *Dr1d* cKO mice were found in the number of marbles buried and the score of nestlet building.

We next determined if the deletion of *Drd1* from BGs impacts delay and trace eyeblink conditioning, two types of motor learning that involve the cerebellum and the hippocampus[61]. In the delay eyeblink conditioning, the conditioned stimulus (CS) was paired with the unconditioned stimulus (US), with the CS preceding the US and ending concomitantly with the US. In contrast, in the trace eyeblink conditioning the CS and US are separated by a stimulus-free interval. In both conditioning, control and *Drd1* cKO mice were subjected to a protocol consisting of acquisition, extinction, reacquisition, and retention (Fig. 6e, f). After 12 sessions of pairings mice showed a significant percentage of conditioned responses (CRs). Following the acquisition mice underwent 3 sessions of extinction, where only the CS was presented. The percentage of CRs displayed a rapid decline. In the reacquisition phase, mice quickly reached an asymptotic CR

rate in 3 sessions. To test the long-term retention, mice received CS-US pairings again after a 2-wk break. Mice fast recalled the task, showing high percent CRs within 2 sessions. When percent CRs were compared between control and *Drd1* cKO mice, no difference in either delay or trace eyeblink conditioning was observed in any phase. Moreover, CR amplitude measured by fraction eye closure (FEC) from the CS-alone presentation was also indistinguishable between two groups (Fig. 6e, f and Supplementary Fig. 9a, b). To ask why the eyeblink conditioning is intact in *Drd1* cKO mice, we evaluated cellular activity by performing immunostaining of the immediate early gene *c-fos*, as used previously in eyeblink conditioning[62]. The number of c-fos-positive cells or the intensity of c-fos immunofluorescence serves as an estimate of neuronal activity[63]. Indeed, as the hippocampus is required for conditioning, there was an increased number of c-fos-positive cells in control and *Drd1* cKO mice, compared with naïve mice (Supplementary Fig. 10a, b). As no discrete c-fos-positive cells were identified in the cerebellum, the intensity of c-fos was measured as an alternative metric. In contrast to the hippocampus, we did not find any difference between the control, *Drd1* cKO, and naïve mice after either delay or trace eyeblink conditioning (Supplementary Fig. 10c, d). These data indicate that eyeblink conditioning does not involve strong neuronal activity in the cerebellum and that D1Rs may not have a critical role.

## DAergic modulation of social activity
Apart from motor activity, the cerebellum is known to be involved in social activity and cognitive function[61,64]. To determine whether *Drd1* cKO affects social behaviors, we used a three-chamber interaction arena to test sociability and social memory sequentially. For the sociability test, mice were allowed to freely explore either a chamber containing a novel mouse (S1) restrained under an inverted cup or a chamber containing an empty inverted cup (E). Control mice spent significantly more time interacting with S1 than with E (Fig. 7a, b). However, *Drd1* cKO mice showed no significant preference between

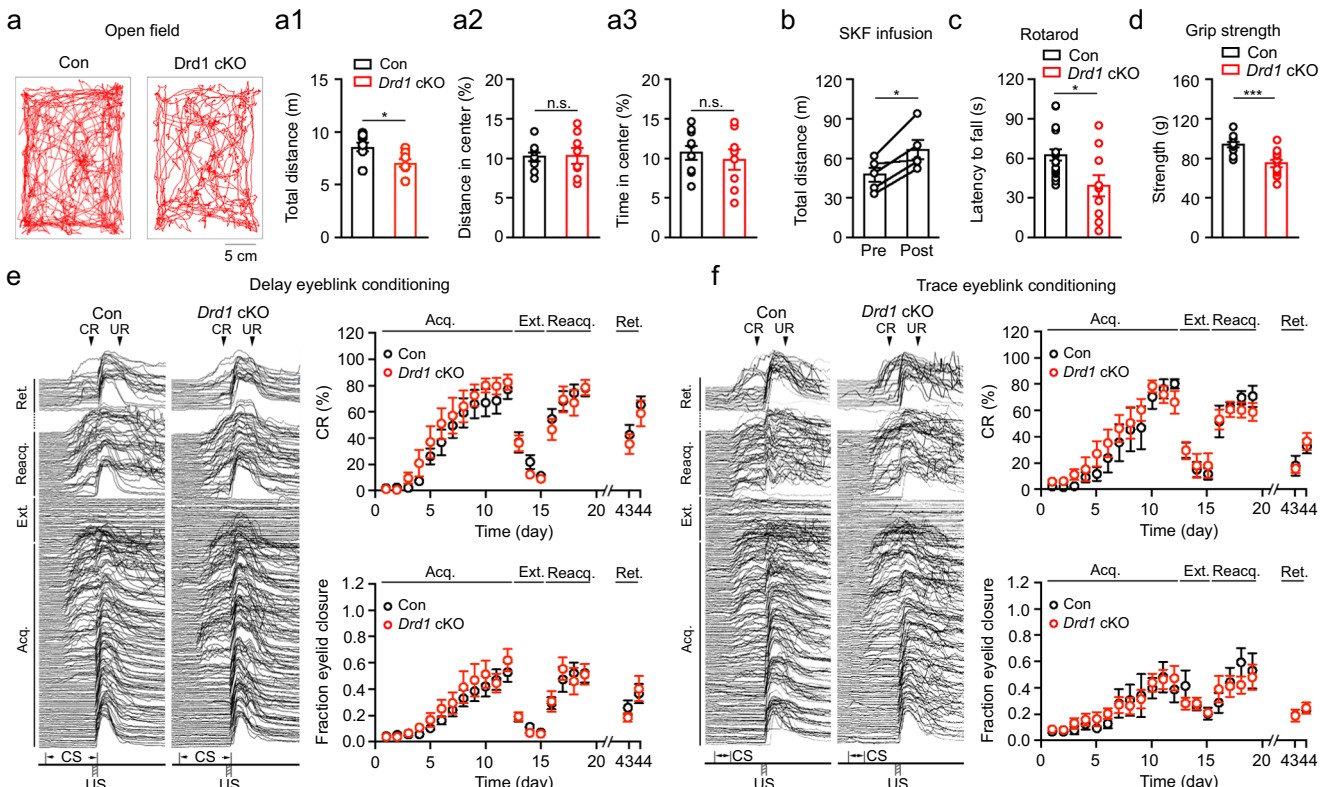

**Fig. 6 | Locomotor activity but not eyeblink conditioning is impaired in *Drd1* cKO mice. a** Total distance traveled in an open filed arena (20 × 30 cm) for 10 min by control (*n* = 10) and *Drd1* cKO (*n* = 8) mice (a1). Distance traveled in the center zone over the total distance was calculated for percent distance in center (a2); time spent in the center zone over the total time was calculated for percent time in center (a3). Data are presented as mean ± SEM. Unpaired two-sided Student's *t* test (*p* = 0.0311, Total distance; *p* = 0.8940, Distance in center; *p* = 0.5591, Time in center; Con vs. *Drd1* cKO). **b** Total distance traveled in an open field arena (60 × 60 cm) for 30 min by mice receiving SKF83822 microinfusion for 5 days (*n* = 5 mice). Data are presented as mean ± SEM. Paired two-sided Student's *t* test (*p* = 0.0162, Con vs. *Drd1* cKO). **c** Latency to fall on a rotarod in control (*n* = 17 mice) and *Drd1* cKO (*n* = 10 mice) groups. Data are presented as mean ± SEM. Unpaired two-sided

Student's *t* test (*p* = 0.0113, Con vs. *Drd1* cKO). **d** Front grip strength measured in control (n = 11 mice) and *Drd1* cKO (n = 13 mice) groups. Data are presented as mean ± SEM. Unpaired two-sided Student's *t* test (*p* = 0.0005, Con vs. *Drd1* cKO). **e**, **f** Percent CR (from CS-US paring) and fraction eye closure (from CS only) during acquisition, extinction, reacquisition, and retention phases of delay (**e**; *n* = 12 mice, Con; *n* = 10 mice, *Drd1* cKO) and trace eyeblink conditioning (**f**; *n* = 5 mice, Con; *n* = 9 mice, *Drd1* cKO). Left, eyeblink traces from one control mouse and one *Drd1* cKO mouse. One of 10 CS-US pairs in each session was plotted. CS, conditioned stimulus; US, unconditioned stimulus; CR, conditioned response; UR, unconditioned response. Data are presented as mean ± SEM. Repeated measures two-way ANOVA (*p* = 0.7108, Delay eyeblink conditioning; *p* = 0.6071, Trace eyeblink conditioning; Con vs. *Drd1* cKO). n.s., not significant.

the S1 and E. As locomotor activity may affect the number of contacts, we calculated the preference index by normalizing the time difference to the total time (Supplementary Fig. 11). The preference index of interaction time in control was statistically different than chance (one-sample *t* test against chance *p* < 0.05), and higher than *Drd1* cKO mice. Immediately following the sociability test, we placed a second novel mouse (S2) under the previously empty cup and allowed the test mice to explore both chambers. Indicative of social memory for S1 and their preference for S2, control mice spent more time interacting with S2 (Fig. 7a, b and Supplementary Fig. 11). However, *Drd1* cKO did not show a preference between S1 and S2. These data suggest that *Drd1* cKO mice demonstrate impaired social interaction and social memory.

Levels of mammalian target of rapamycin (mTOR) signaling are known to be positively correlated with impaired social interaction, and treatment with the mTOR inhibitor rapamycin improves social interaction deficits[65]. Consistently, in *Drd1* cKO mice with impaired social activity, phosphorylation levels of mTOR were significantly higher (Fig. 7c). We also evaluated levels of phosphorylated mTOR in slices treated with SKF83822 (Fig. 7d). SKF83822 treatment resulted in a decrease in p-mTOR levels, which were prevented by pretreatment with SCH23390. These data suggest that DAergic activity in the cerebellum is critically engaged in social behavior by acting through mTOR signaling.

## Discussion

Our study demonstrates the expression of DA, D1R, and DAT in the cerebellum, suggesting an intrinsic cerebellar DAergic system. Previous studies reported that D1Rs are expressed in cerebellar ML and/or PCL[23,66–68]. Here, we further demonstrate that D1Rs are expressed in BGs while DAT is in both BGs and PCs. Astrocytes were previously found to express several DA receptor types[69–71]. Pharmacological activation of D1Rs with their agonists activates PKA and ERK1/2 in cultured striatal astrocytes to promote astrocyte migration[69,72]. Stimulation of D1Rs also triggers Ca$^{2+}$ transients in astrocytes depending on the NAD + /NADH redox state[73,74]. By exploiting the pharmacological approach and DA sensor imaging, we provide solid evidence that DA can be synthesized in PCs by the enzyme CYP2D. Several studies demonstrate that human CYP2D6 or rodent CYP2Ds can metabolize *m*-tyramine and *p*-tyramine into DA[37–40]. More notably, CYP2D6 activity is associated with the incidence and prevalence of brain disorders. In PD patients, CYP2D6 protein levels are largely reduced in the cerebellum of PD patients;[75] CYP2D6 polymorphism leads to a higher susceptibility to PD[76]. Individuals with CYP2D6 variants also present personality, cognition, and psychopathological vulnerability[77]. Our identification of CYP2D for DA synthesis does not entirely exclude the possibility of the production of DA by TH. Modest TH expression was observed in PCs in the vermal lobules V and VI[78]. Axonal fibers derived from the VTA were sparsely found in the deep cerebellar nuclei or in crus 1, and 2;[79] fibers

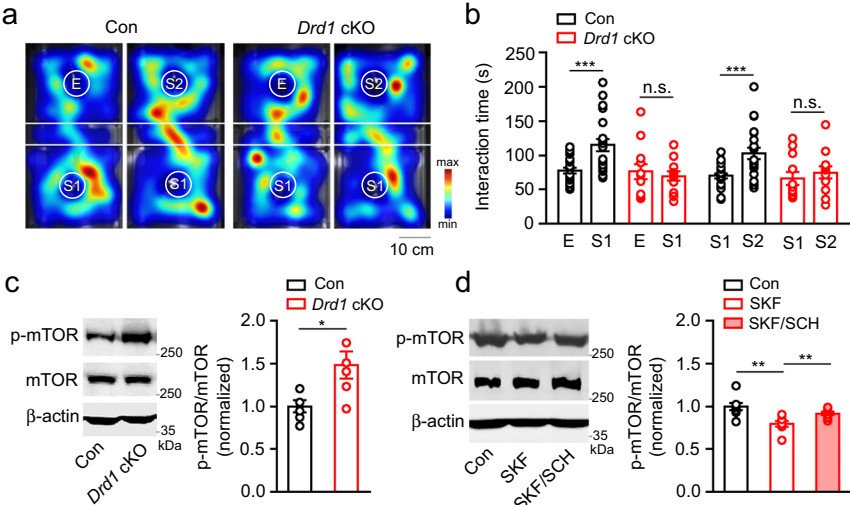

**Fig. 7 | DAergic modulation of social behavior. a** Heat map of control and *D1R* cKO mice encountering the empty inverted cup (E) or the cup containing the first novel mouse 1 (S1) or S2 during the three-chamber social test. Minimal and maximum values indicate the relative accumulating activity in each area of the arena. **b** Time spent in the interaction with E, S1, or S2 (*n* = 22 mice, Con: *n* = 12 mice, *Drd1* cKO). Data are presented as mean ± SEM. Unpaired two-sided Student's *t* test (*p* = 0.0005, Con E vs. S1; *p* = 0.6135, *Drd1* cKO E vs. S; *p* = 0.0010, Con S1 vs. S2; *p* = 0.4996, *Drd1* cKO S1 vs. S2). **c** Phosphorylation levels of mTOR in the cerebellum

of control and *Drd1* cKO mice (*n* = 6, each group). Data are presented as mean ± SEM. Mann–Whitney two-sided test (*p* = 0.0019, Con vs. *Drd1* cKO). **d** Phosphorylation levels of mTOR in cerebellar slices treated with vehicle (DMSO) for 1 h, SKF83822 for 1 h, or pretreated with SCH23390 followed by SKF83822 for 1 h (*n* = 8, each group). Data are presented as mean ± SEM. Multiple unpaired two-sided Student's *t* test (*p* = 0.0020, Con vs. SKF, *p* = 0.0096, SKF vs. SCH). n.s., not significant.

labeled with anti-TH were shown unevenly in the ML, PCL, and GL of a few lobules, such as V and VI[80,81]. All these findings indicate low TH expression in a limited number of lobules, which is consistent with our result from TH immunolabeling. Conversely, CYP2D is expressed in all lobules throughout the cerebellum, commensurate with D1R expression. We further demonstrate that DA detected by its sensor was abolished by in vitro or in vivo treatments of the CYP2D inhibitor. These data strongly suggest that CYP2D in PCs is the major enzyme required for DA synthesis in the cerebellum. This finding is also congruent with a previous report showing that strong depolarization in PCs can trigger vesicular DA from PCs[23]. Nevertheless, these immunostaining and pharmacological results must be further substantiated in mice lacking CYP2D specifically in the cerebellum. Furthermore, to confirm the direct action of released DA on BGs, for our future experiments it is necessary to use viral vectors that express DA sensors specifically in BGs.

Our data show that D1R activation depolarizes membrane potential and induces intracellular Ca²⁺ signals. BGs express high a density of inwardly rectifying K⁺ channels, which are critical for maintaining glial membrane potentials[82]. BG depolarization could be caused by D1R-mediated inhibition of inwardly rectifying K⁺ channels (Kir), as seen in striatal neurons[83]. Depolarization of membrane potential can be a contributing factor to the fast rises of Ca²⁺ signals. In addition to these rapid responses, D1R treatment also yields long-term cellular modification by assisting the insertion of GluA1 into the BG membrane. This GluA1 membrane trafficking involves PKA activity and phosphorylation of GluA1 at Ser845, similar to that observed in neurons[58,59]. The increase of GluA2-lacking CP-AMPARs, resulting from GluA1 insertion, would prolong membrane depolarization and Ca²⁺ influx. Fast Ca²⁺ rises from the internal store and slow rises mediated by CP-AMPARs are expected to affect PC activity and behavior significantly. The D1R-mediated signaling is generally thought to be associated with Gs-coupled receptor stimulation[84]. Our data indeed show that D1R activation promotes GluA1 insertion, which can be blocked by inhibiting protein kinase A (PKA), a downstream signal of Gs. Furthermore, our pharmacological experiments also reveal that D1Rs could induce Ca²⁺ transients by signaling through Gq-coupled

receptors and their downstream effector IP3Rs. This effect can be mimicked by the chemogenetic stimulation of excitatory hM3DGq DREADD receptors. Consistently, one study showed that 2-APB treatment could block D1R-induced IP3R signaling[85]. More notably, evidence demonstrates that D1R also stimulates Gq in astrocytes[74,86]. Together with these and other previous reports[87–91], we speculate that D1R-mediated signaling can act through both Gs and Gq, which work separately or together in triggering Ca²⁺ rises from different sources in BGs.

BG activity is long known to impact PC synaptic transmission and plasticity[26]. Depolarization of a single BG induces a significant increase in the frequency of sEPSCs recorded in an adjacent PC, which was believed to be caused by modulation of presynaptic glutamate release[92]. Increasing Ca²⁺ signals in BGs results in a decrease in extracellular K⁺, thereby altering PC membrane potential and transiently increasing spike activity[93].

In agreement with these previous reports, the D1R agonist that can depolarize BGs induces an initial increase in sEPSC frequency in PCs. This enhancement, however, is transient; a persistent reduction ensues after extended exposure. We are uncertain how depolarized membrane potential and increased Ca²⁺ signals in BGs, mechanistically, lead to these alterations in PCs. Considering the rapid action and the existence of presynaptic glutamate receptor (GluR) distribution, we postulate that the increased sEPSC frequency may be caused by activation of presynaptic ionotropic GluRs[94,95] and that the subsequent decrease may be due to a strong delayed inhibition of presynaptic metabotropic GluR-mediated transmitter release[96,97]. Our data further demonstrate that the inhibitory input is upregulated in PCs, suggesting that neurotransmitters (e.g., glutamate) released from BGs activate both GABAergic MLIs and PC presynaptic sites and result in reduced overall neurotransmission at both PF-PC and CF-PC synapses. Toning down synaptic transmission leads to decreased PC excitability, which may serve as a common mechanism underlying cerebellum-related behaviors[98]. In the hippocampus, a similar circuit inhibition is mediated by DA's action on the cortical input[99,100].

Ca²⁺ rise in BGs is associated with locomotion[101,102]. Consistently, our data show that microinfusion of the D1 agonist in the cerebellum

augments locomotor activity, while *Drd1* cKO mice exhibit decreased locomotor activity. We did not see an impairment in cerebellum-related delay or trace eyeblink conditioning. Immunostaining for c-fos did not show a significant increase in intensity after eyeblink conditioning in either control or *Drd1* cKO mice, suggesting BGs or PCs are not strongly activated. Whether the deep cerebellar nuclei or cerebellar cortex is a hub for eyeblink conditioning is still under debate[103]. Our findings suggest that the cerebellar cortex might have a less important role in eyeblink conditioning. However, it is also possible that c-fos immunostaining may not be an ideal tool to detect accumulating cell activity if these cells like PCs and BGs are intrinsically active. In comparison with the importance of BGs in social interaction, we speculate that in BGs, the subcellular and molecular signaling required for eyeblink conditioning and other cerebellar functions may be profoundly different. As the cerebellar cortex is critical for the timing of the conditioning[104], we also cannot exclude the possibility that delay or trace eyeblink conditioning with other intervals may display the deficit. Collectively, these findings suggest that the DA pathway in the cerebellum is involved in social activity but likely not in classical conditioning.

In summary, our results reveal the molecular, cellular, and circuit mechanisms underlying the DA system in the cerebellum (Fig. 8). We show that PCs synthesize and release DA in an activity-dependent manner. The released DA binds D1Rs in BGs and then promotes membrane depolarization, $Ca^{2+}$ signaling, and membrane insertion of AMPAR GluA1 subunits. These actions likely trigger glutamate release from BGs, which then enhances interneuron activity and reduces PF- and CF-PC synaptic transmission. The altered synaptic input to PCs modulates their firing frequency and pattern, ultimately impacting locomotor and social behaviors. These findings indicate that the cerebellar DAergic system has a critical pathophysiological role in many disorders associated with motor and social dysfunction.

## Methods

### Animal models

Animals were handled and housed according to the Committee on Laboratory Animal Resources of the National Institutes of Health. Both male and female mice at age of 2–3 months were used in our study. Mice were housed at the room temperature (25 °C) with a 12-h light/dark cycle and humidity between 40 and 60%. Mice were provided ad libitum access to food and water. All experimental protocols were reviewed and approved annually by the Institutional Animals Care and Use Committee of the University of Alabama at Birmingham (IACUC-22247). Previous reports have described the generation and genotyping of the following mouse lines: *Drd1*-tdTomato[33], *Drd2*-EGFP[34], *Slc6a3*-Cre[105], *Pcp2*-Cre[106], *Slc1a3*-Cre[107], and *Drd1tm2.1*[108]. *Drd1a*-tdTomato mice express the fluorescent tdTomato in D1R-expressing cells, while *Drd2*-EGFP mice express the fluorescent EGFP in D2R-exprssing cells. *Slc6a3*-Cre, *Pcp2*-Cre, and *Slc1a3*-Cre mice express Cre recombinase in DAergic neurons, most PCs, and glia, respectively. We followed the genotyping and husbandry protocols provided by the vendors. For Cre-expressing mice, the heterozygous offspring were used. To generate *Drd1* cKO mice, *Slc1a3*-Cre mice were crossed to *Drd1tm2.1* mice and tamoxifen (T5648, Sigma-Aldrich) prepared in corn oil was administered (i.p., 100 mg/kg) for 5 consecutive days to their offspring starting at postnatal day 30 (PND 30). Homozygous *Drd1* cKO mice were used for experiments at age of 3 months.

### Stereotactic injections

All adeno-associated viruses (AAVs) were obtained from Addgene and delivered via stereotactic intracranial injections. Mice at PND 20-27 were anesthetized with 4% isoflurane in 100% oxygen gas; anesthesia was maintained with 1-2.5% isoflurane. Mice were placed in a stereotactic frame (David Kopf Instruments), and their body temperature was maintained with a heating pad. The scalp was shaved and then sterilized with 70% ethanol. A rostral-caudal incision was made to

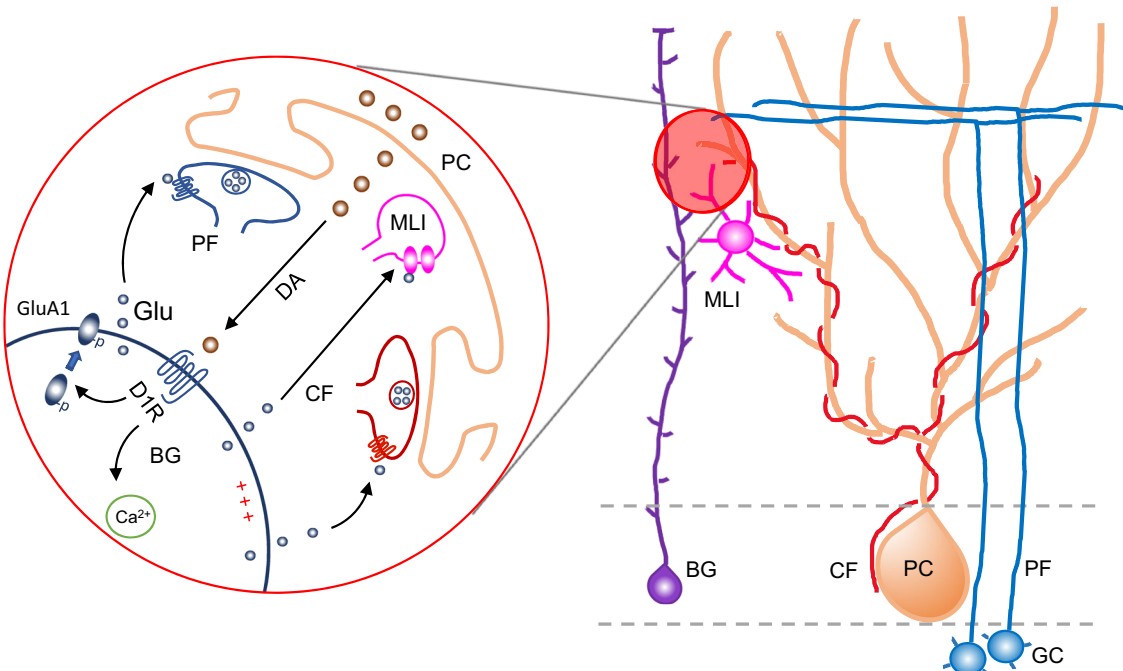

**Fig. 8 | Diagram of the action of the DA system in the cerebellum.** Our findings suggest a model in which PCs synthesize DA through a non-canonical pathway and secrete it in an activity-dependent manner. The released DA binds D1Rs in BGs, inducing membrane depolarization and $Ca^{2+}$ signaling and driving membrane insertion of AMPAR GluA1 subunits. These actions may trigger glutamate release from BGs, which then enhances interneuron activity and reduces PF- and CF-PC synaptic transmission by potentially activating presynaptic GluRs. The increased inhibitory and the reduced excitatory inputs to PCs alter their firing frequency and pattern, ultimately impacting locomotor and social behaviors. BG, Bergmann glial cell; GC, granule cell; PC, Purkinje cell; MLI, molecular layer interneuron; CF, climbing fiber; PF, parallel fiber; DA, dopamine; D1R, D1 dopamine receptor; Glu, glutamate.

access the skull, a hole was drilled, and virus (0.5 µl at a titer of 0.5-1×10$^{13}$ GC/ml) was delivered through a 2.5 µl syringe (Hamilton Company) at a rate of 0.25 µl/min using a microsyringe pump (UMP3 UltraMicroPump, Micro4, World Precision Instruments). For expression of the DA sensor, Ca$^{2+}$ indicator, and excitatory DREADD, pAAV5-CAG-dLight1.1, pZac2.1-gfaABC1D-cyto-GCaMP6f, and pAAV-GFAP-hM3D(Gq)-mCherry was injected into the cerebellum, respectively (AP = −6.75 mm, ML = −1.8 mm, DV = −1.2 mm). For expression of the DA sensor in the striatum, pAAV5-CAG-dLight1.1 was injected (AP = 0.7 mm, ML = −1.75 mm, DV = −2.9 mm). For expression of YFP in DAergic neurons, pAAV5-Ef1a-DIO EYFP was injected into the VTA and SNc (AP = −3.0 mm, ML = ±1.3 mm, DV = −4.1 mm), or the LC (AP = −5.52 mm, ML = ±0.8 mm, DV = −3.0 mm) of *Slc6a3*-Cre mice. Following the injections, the incision was closed with surgical glue. Topical antibiotic ointment (bacitracin zinc, neomycin sulfate, and polymyxin B sulfates; Actavis) was applied to the incision, and carprofen (i.p. 5 mg/kg; Zoetis) was administered. Mice were used for experiments 3-4 wks later.

For identification of the possible DAergic input from the SNc to the cerebellum, mice at PND 40 were injected with Dextran-Alexa Fluor 488 (500 nl, Thermo Fisher Scientific) in the SNc and retrobeads (200 nl, Lumafluor) in the cerebellar crus 1 and 2. Mice were used for experiments 4 d later.

### In vitro cerebellar slices

Mice at 2-3 months were deeply anesthetized with a ketamine and xylazine mixture, and transcardially perfused with ice-cold cutting artificial cerebrospinal fluid (aCSF) containing (in mM): 87 NaCl, 2.5 KCl, 0.5 CaCl$_2$, 7 MgCl$_2$, 1.25 NaH$_2$PO$_4$, 25 NaHCO$_3$, 25 glucose, and 75 sucrose, bubbled with 95% O$_2$/5% CO$_2$. The brain was rapidly removed and cut transversely at 300 µm using a vibratome (VT1200S, Leica Microsystems). Slices were transferred to normal aCSF containing (in mM): 119 NaCl, 2.5 KCl, 2.5 CaCl$_2$, 1.3 MgCl$_2$, 1.3 NaH$_2$PO4, 26 NaHCO$_3$, and 20 glucose, at 32 °C for 30 min and then allowed to recover for 1 h at room temperature before recordings.

### Whole-cell patch clamp electrophysiology

Individual slices were transferred to a submerged chamber mounted on a fixed-stage upright microscope (Axio Examiner.D1, Zeiss) and continuously perfused with normal oxygenated aCSF at room temperature. Signals in BGs or PCs were acquired with a MultiClamp 700B amplifier (Molecular Devices), filtered at 2 kHz, and digitized at 10 kHz with ITC-18 A/D-D/A interfaces (Instrutech)[109]. Input resistance was measured with hyperpolarizing voltage pulses (50 ms, 20 mV). Cells with series resistances above 25 MΩ were discarded, and cells were also excluded if any whole-cell parameter (i.e. Cm, Ri, Rs) changed by ≥ 20% during the recordings.

For current or voltage clamps in BGs, the intracellular solution contained (in mM): 135 K-gluconate, 10 KCl, 10 HEPES, 1 MgCl$_2$, 2 Mg-ATP, 0.3 Na-GTP (290-300 mOsm, pH 7.3, 4-5 MΩ). To evaluate the effect of D1R activation on cell-intrinsic property, BGs were recorded at resting membrane potential (RMP) and membrane potential was monitored in slices treated with SKF83822 (10 µM, Tocris). For slices expressing hM3D(Gq), membrane potential in BGs was recorded during CNO (1 µM, Tocris) treatment. To test inward rectification in BGs, *S*-AMPA (10 µM, 20-ms duration, Tocris) was pressure puffed to slices in the presence of the GABA$_A$R antagonist picrotoxin (50 µM, Tocris), the NMDAR antagonist D-AP5 (50 µM, Tocris), and cyclothiazide (CTZ; 50 µM, Tocris). The internal solution was supplemented with spermine (100 µM, Tocris). NASPM (50 µM, Tocris) was used to selectively inhibit the activity of GluA2-lacking AMPARs; the AMPAR antagonist CBQX (20 µM, Tocris) was used to completely block all AMPAR channels.

To evaluate synaptic input to PCs, PFs and CFs were alternately stimulated with a 10-s interval by using aCSF-filled pipettes connected to an isolated stimulator (ISO-Flex, AMPI); PSCs were recorded in PCs

held at −60 mV under the voltage-clamp mode. The intracellular solution contained (mM): 120 Cs-gluconate, 17.5 CsCl, 10 Na-HEPES, 4 Mg-ATP, 0.4 Na-GTP, 10 Na$_2$-creatine phosphate, 0.2 Na-EGTA (290-300 mOsm, pH 7.3, 3-4 MΩ). To record PSCs without the addition of the Na$^+$ channel blocker QX314 (5 mM, Tocris) to the intracellular solution, stimulus intensity was adjusted to ensure that EPSCs were clamped. Following the stable baseline recording, SKF83822 was applied to slices in normal aCSF. The amplitude of PSCs was analyzed during SKF83822 treatment. In slices expressing hM3D(Gq), CNO (1 µM) was perfused to slices. PC spikes were then recorded at near firing threshold under the current-clamp mode, or evoked PSCs were recorded at PF-PC synapses under the voltage-clamp mode. To record spontaneous spiking in PCs, aCSF-filled pipettes formed a loose seal on PCs and action potentials were recorded under the cell-attached mode. The number of spikes was binned each min and compared during SKF83822 treatment. The coefficient of variation (CV) 2 was defined as $\frac{2 \times |ISI_{i+1} - ISI_i|}{ISI_{i+1} + ISI_i}$, where $ISI_i$ indicates the $i$th inter-spike interval. The average CV2 of all consecutive spikes was calculated for a neuron.

Spontaneous EPSCs (sEPSCs) in PCs were recorded with membrane voltage held at −60 mV in the presence of picrotoxin. Spontaneous IPSCs (sIPSCs) in PCs were recorded at −60 mV in the presence of NBQX (20 µM) and D-AP5 (50 µM). The intracellular solution for sEPSC recordings is described as above. The intracellular solution for sIPSC recordings contained (in mM): 140 CsCl, 2 MgCl$_2$, 10 HEPES, 5 EGTA, 4 Mg-ATP, 0.4 Na-GTP, 0.5 CaCl$_2$. sEPSCs and sIPSCs were analyzed using the MiniAnalysis program (Synaptosoft) with the detection threshold set at 8 pA. Their frequency and amplitude were binned and then analyzed.

### Ca$^{2+}$ and DA sensor imaging

For Ca$^{2+}$ imaging with the organic Ca$^{2+}$ indicators, Fluo 5F (100 µM, Thermo Fisher Scientific) or OGB-1 (200 µM, Thermo Fisher Scientific) was included in the intracellular solution and imaged in BGs 15 min after whole-cell access. For the Fluo 5F imaging, Alexa Fluor 594 (100 µM, Thermo Fisher Scientific) was also dissolved in the solution and introduced into the cell. Ca$^{2+}$ imaging with the genetically encoded Ca$^{2+}$ indicator was performed in BGs expressing AAV5-gfaABC1D-cyto-GCaMP6f. Fluo 5 F, OGB-1, GCaMP6f was excited by a laser-LED hybrid source using less than 10% of full power (460-495 nm, X-Cite Turbo) and signals were detected with a quantitative electron-multiplying CCD camera (QuantEM:5125 C, Photometrics). For fast recordings of Ca$^{2+}$ signals, images were acquired with 60-ms exposures under continuous light excitation, whereas for recording of slow spontaneous signals, images were captured at 1 frame/s using a light-dark discontinuous mode. Fluo 5F imaging was performed in control BGs held at −80 mV, or BGs depolarized to −20 mV or hyperpolarized to −120 mV three times at 0.1 Hz with 1-s duration each, and Ca$^{2+}$ signals were evaluated during the 10-min recordings. OGB-1 imaging was performed in BGs in combination with whole-cell recordings of membrane of potentials, and Ca$^{2+}$ signals induced by local application of AMPA were evaluated during treatments of NASPM and CNQX. These Ca$^{2+}$ signals were measured within regions of interest (ROIs) defined over the proximal region of BG soma after the background fluorescence intensities had been subtracted; fluorescence bleaching was corrected by exponential fit subtraction. Ca$^{2+}$ signals obtained via GCaMP6f imaging in BGs were analyzed using AQuA software as described previously[110]. To examine the role of D1R activation in BG Ca$^{2+}$ signaling, SKF83822, SKF83822 in the presence of 2-APB (50 µM, Tocris), or SCH23390 (10 µM, Tocris) was applied during imaging. In cerebellar slices expressing both GCaMP6f and hM3D(Gq), CNO was perfused to activate hM3D(Gq). Ca$^{2+}$ frequency or peak ΔF/F was compared before (2-min duration) with after current injections or pharmacological treatments (8-min duration).

DA sensor imaging was performed in slices expressing AAV5-CAG-dLight1.1[42]. DA was applied at different concentrations (0.5, 2, and 8 µM)

to control slices or slices pretreated with SCH23390 (10 μM). dLight1.1 imaging was also conducted in mice or slices treated with fluoxetine, quinidine, or reserpine. Background-subtracted fluorescence intensity was measured within randomly selected ROIs in cerebellar molecular layer, and the amplitude of dLight1.1 intensity was calculated.

## Immunostaining

Mice were anesthetized and transcardially perfused with 4% paraformaldehyde in PBS. Brains were removed and postfixed overnight at 4 °C. Sagittal or coronal brain sections were cut at 60 μm using a vibratome. Sections were permeabilized with 0.25% Triton X-100 for 2 h and blocked with 10% normal goat serum for 1 h. Sections were incubated at 4 °C overnight with blocking solution containing the following primary antibodies: mouse anti-GFAP (1:500, Sigma-Aldrich), rabbit anti-DsRed (1:2000, TaKaRa), mouse anti-RFP (1:1000, Rockland Immunochemicals), mouse anti-D1R (1:200, Sigma-Aldrich), rabbit anti-D1R (1:200, Abcam), mouse anti-DAT (1:200, Thermo Fisher Scientific), rabbit anti-GFP (1:2000, Abcam), chicken anti-GFP (1:2000, Abcam), mouse anti-calbindin D28K (CB) (1:500, Santa Cruz Biotechnology), rabbit anti-N terminal CYP2D6 (1:500; AV41675, Sigma-Aldrich), rabbit anti-C terminal CYP2D6 (1:500; PA5-79129, Thermo Fisher Scientific), mouse anti-TH (1:500, Sigma-Aldrich), rabbit anti-mCherry (1:2000, Abcam), rabbit anti-S100β (1:800, Sigma-Aldrich), guinea pig anti-c-fos (1:800, Synaptic Systems), or guinea pig anti-PV (1:800, Synaptic Systems). After the primary antibodies, sections were rinsed with PBS three times each 10 min and incubated for 2 h at room temperature with the secondary antibodies, including Alexa Fluor 488 goat anti-mouse, Alexa Fluor 488 goat anti-rabbit, Alexa Fluor 488 goat anti-chicken, Alexa Fluor 488 goat anti-guinea pig, Alexa Fluor 594 goat anti-mouse, Alexa Fluor 594 goat anti-rabbit (1:1000, Jackson ImmunoResearch Laboratories). Sections were coverslipped with Vectashield mounting media (Vector Laboratories). All images were acquired using ×10 (0.3 NA), ×20 (0.8 NA), or ×63 (1.4 NA) objectives in an LSM-800 Airyscan confocal microscope (Zeiss). For counting the number of puncta and measuring the fluorescence intensity of immunostaining, images were acquired with the same settings of laser power and PMT voltage and gain for semi-quantitative comparison using ImageJ (NIH). The number of puncta was counted by unbiased ImageJ particle analysis with defined puncta size range.

To identify the distribution of AMPAR subunits in the cerebellum, BGs or PCs in acute slices were filled with biocytin (8 mM, Sigma-Aldrich) during whole-cell recordings. To minimize biocytin diffusion among BGs, slices were pretreated with the gap junction blocker carbenoxolone (50 μM, Tocris) to minimize biocytin diffusion. Following the fixation, permeabilization, and blocking, slices were incubated with rabbit anti-GluA1(1:500, Sigma-Aldrich), rabbit anti-GluA2 (1:500, Thermo Fisher Scientific), mouse anti-GluA2 (1:500, Thermo Fisher Scientific), or rabbit anti-GluA4 (1:500, Sigma-Aldrich). Slices were then incubated with streptavidin-conjugated Alexa Fluor 488 (1:200, Thermo Fisher Scientific) and Alexa Fluor 594 goat anti-mouse or anti-rabbit secondary antibodies. z-stack confocal images of BGs or PCs and GluA1, GluA2, or GluA4 immunostaining were acquired using LSM-800 Airyscan confocal microscope and their spatial relationship was evaluated in three dimensions. To identify the distribution of D1Rs in BGs, BGs were filled with biocytin and immunostained with anti-DsRed in Drd1a-tdTomato slices or anti-D1R in control slices. To identify the path of CFs to PCs, the anterograde tracer Dextran-Alexa Fluor 594 (500 nl, Thermo Fisher Scientific) was injected into the inferior olive and allowed 4 days for labeling CF afferents in the cerebellum, followed by immunostaining with CB for labeling PCs.

## Single-molecule RNA FISH

Mice were transcardially perfused with 4% paraformaldehyde in PBS. Brains were removed and postfixed overnight at 4 °C. Sagittal sections were cut at 30 μm. Sections were washed three times with PBS and

permeabilized in 70% ethanol at 4 °C for 5 min. Sections were then washed in Stellaris Wash buffer A for 5 min at room temperature. Sections were transferred into hybridization buffer containing Dr1d mRNA probes (0.5 nM) and incubated for 15 h at 37 °C with gentle rocking. Sections were washed three times in Stellaris Wash buffer A at 37 °C, and then in Stellaris Wash buffer B for 5 min. Sections were mounted and coverslipped. All images were acquired using LSM-800 Airyscan confocal microscope with a 20× (0.8 NA) objective.

## Western immunoblotting

Fresh whole cerebellar samples or acute slices were homogenized in NP-40 buffer (20 mM Tris, pH 8.0, 137 mM NaCl, 10% glycerol, 1% Nonidet P-40, 2 mM EDTA) containing protease and phosphatase inhibitor cocktail (Thermo Fisher Scientific). The homogenates were maintained with constant agitation for 2 h at 4 °C and centrifuged at 10,000 g for 20 min. The supernatants were aspirated and protein concentrations determined by the Lowry method. Equal amounts of protein sample were denatured in loading buffer (125 mM Tris at pH 6.8, 20% glycerol, 6% SDS, and 5% 2-mercaptoethanol), boiled for 3 min, and subjected to SDS-PAGE. Proteins were transferred to PVDF membrane and blocked with 5% nonfat milk in TBST (20 mM Tris, pH 7.6, 150 mM NaCl, and 0.1% Tween-20) for 1 h. Membranes were incubated with rabbit anti-GluA1(1:500, Sigma-Aldrich), rabbit anti-GluA2 (1:500, Thermo Fisher Scientific), mouse anti-GluA2 (1:500, Thermo Fisher Scientific), rabbit anti-GluA4 (1:500, Sigma-Aldrich), rabbit anti-p831 GluA1 (1:200, Sigma-Aldrich), rabbit anti-p845 GluA1 (1:200, Thermo Fisher Scientific), rabbit anti-mTOR (1:800, Cell Signaling Technology), or rabbit anti-p2448 mTOR (1:500, Cell Signaling Technology). After rinsed with PBS three times each 10 min, the membrane was incubated for 2 h at room temperature with corresponding peroxidase-conjugated secondary antibodies (1:2000, Jackson ImmunoResearch Laboratories). The proteins were detected using the Pierce ECL Substrate (Thermo Fisher Scientific), and signals were captured on autoradiography film and quantified by computer-assisted densitometry. Membranes were re-probed for the loading control with β-actin (1:2000, Thermo Fisher Scientific) and detected with IRDye 800CW goat anti-mouse using Odyssey infrared imaging system (Li-Cor Bioscience).

## Surface biotinylation

Slices were rinsed with ice-cold PBS for 30 min and then incubated with Sulfo-NHS-SS-biotin (1 mg/ml, Thermo Scientific) at 4 °C for 45 min. Unreacted biotinylation reagents were quenched in buffer containing 192 mM glycine, 25 mM Tris, pH 8.3). Slices were lysed in NP-40 lysis buffer and protein concentration of each lysate was quantified. Equal amounts of protein lysates were incubated overnight with streptavidin agarose beads (Thermo Scientific). Beads were then washed three times with ice-cold lysis buffer, and biotinylated protein were eluted with sample buffer. Surface or total proteins were then subjected to Western immunoblotting.

## Fluoxetine, quinidine, or reserpine treatment

In vitro fluoxetine or quinidine treatment was performed in acute cerebellar slices that expressed dLight1.1. Slices were submerged in a 12-well plate connected to a custom-built perfusion system, and perfused with fluoxetine (100 μM) or quinidine (50 μM) for 2 h at a rate of 3 ml/min. In vitro reserpine (1 μM) treatment was performed in slices for 10 min. In vivo fluoxetine treatment was performed by administering fluoxetine (i.p., 10 mg/kg) for 2 wks to mice that expressed dLight1.1 in the cerebellum and acute cerebellar slices were then prepared. Control groups received vehicle (DMSO) perfusion or injection. Imaging of dLight1.1 was then performed to measure DA levels.

## SKF83822 microinfusion

Mice were anesthetized and then placed in a stereotactic frame. Cannulae were bilaterally built in the cerebellum (AP = −6.75 mm, ML =

±1.8 mm, DV = -1.2 mm) and secured using screws and C&B Metabond cement (Parkell). Mice were allowed to recovery for 7 d. Microinfusion of SKF83822 (5 mM, 1 µl) into the bilateral cerebellum were performed at a rate of 0.1 µl/min by using a microsyringe pump connected to an internal cannula. Control mice were subject to the same surgery procedure and injected with saline. Microinfusion was performed for 5 consecutive days, and behavioral tests were conducted 1 h after the microinfusion on the last day.

## In vivo fiber photometry

After viral injection, a 2-mm fiber optic cannula (Neurophotometrics or RWD) was implanted at the same coordination as the one for viral injections and affixed to the skull surface using C&B Metabond. After at least 14 days of dLight1.1 expression, fiber photometry data was acquired using a commercial system (FP3002, Neurophotometrics) controlled by an open-source program (Bonsai). The system delivered 470-nm (emit) and 415-nm LED (control), each having a light intensity of around 40 mW, to excite the sensor and the emitted fluorescent signal was collected at 16 frames per second (FPS). During the recording, freely moving animals were put in an open cage (26 cm × 15 cm × 14 cm) for 5 min after acclimation. Fiber photometry data collected from the system were analyzed using published MATLAB script[111].

## Open field test

*Drd1* cKO (*Slc1a3*-Cre[+/−];*Drd1tm2.1*[+/+], with tamoxifen), or control (*Slc1a3*-Cre[+/−];*Drd1tm2.1*[+/+], without tamoxifen) were placed in the center of 60 × 60 or 30 × 40 open arenas. Following 10-min habituation, mice were recorded for 10 or 30 min with IR-sensitive Gigabit Ethernet video camera (ace acA780-75gm, Basler). The arena floor was virtually divided into 9 zones. The recorded activity was analyzed for the total distance, the amount of time spent in the central zone over the total time, or the distance traveled in the central zone over the total distance. These analyses were performed by using video tracking software (Ethovision XT 16, Noldus).

## Rotarod test

*Drd1* cKO or control mice were trained for 3 consecutive days on a rotarod apparatus (ROTOR-ROD, San Diego Instruments). Mice were placed on the rotating rod for a maximum of 1 min and returned to their home cage between trials. Mice were given 4 trials per day with a 10-min inter-trial interval. On the following day, mice were given 3 test trials. The duration spent on the rod was recorded for each trial and the average time was calculated for latency to fall.

## Grip strength test

For forelimb strength test, *Drd1* cKO or control mice were placed on a grip strength apparatus (San Diego Instruments). Mice were lifted by holding their tail and gently pulled backwards until the grip was lost. Mice were given 5 test trials and the peak force of each measurement was recorded. The average value was calculated for analysis.

## Ladder rung walking

To assess walking ability and leg placement, *Drd1* cKO or control mice were tested on horizontal ladder rungs (Bio-FMA, Bioseb). Mice were allowed to move one end to the other end of the ladder rungs. Mouse movement and paw misplacements were detected by two sets of infrared sensors placed above and below the ladder. Mice were given 3 trials with a 10-min inter-trial interval. The average number of footslip errors was calculated for forelimbs and hindlimbs.

## Catwalk

A Catwalk system (CatWalk XT, Noldus) was used to assess gait performance in *Drd1* cKO or control mice. High-speed camera was mounted beneath the glass platform to record walking patterns. Mice were trained for 3 d to travel from one end of the corridor to the other end and were given 3 trials with a 10-min inter-trial interval on the test day. The recorded videos were analyzed for the average stride length of right forepaw (RF), right hindpaw (RH), left forepaw (LF), and left hindpaw (LH).

## Three-chamber social test

*Drd1* cKO or control mice were acclimated to being handled for 3 d. Mice were placed in the center chamber of the arena (60 × 30 × 20 cm) and allowed to freely explore the entire arena for 5 min. Mice were shepherded back to the center chamber and blocked there. A novel mouse was placed under one of the pencil cups located in two side chambers. The blocks were removed, and the test mouse was allowed to explore the chambers for 10 min. After this time, the test mouse was returned to the center chamber and blocked there again. A second novel mouse was put under the previously empty cup. The blocks were lifted, and the test mouse was allowed to explore the chambers for anther 10 min. The amount of time the test mouse spent sniffing either the empty cup or the cup containing sentinel mice was quantified (Ethovision XT 16, Noldus). The preference index was calculated as $\frac{\text{time investigating the cup with the 1st novel mouse} - \text{time investigating the empty cup}}{\text{time investigating the empty cup} + \text{time investigating the cup with the 1st novel mouse}} \times 100$ for the sociability test and $\frac{\text{time investigating the cup with the 2nd novel mouse} - \text{time investigating the cup with the 1st novel mouse}}{\text{time investigating the cup with the 1st novel mouse} + \text{time investigating the cup with the 2nd novel mouse}} \times 100$ for the social memory test.

## Marble burying

Standard polycarbonate cages (40 × 30 × 22 cm) were filled with bedding to a depth of 5 cm. On top of the bedding, 5 rows of 4 marbles were placed and mice were given access to the marbles for 30 min. A marble was scored as buried if 2/3 of its surface area was covered. The total number of buried marbles was counted for each mouse.

## Nestlet building

Mice were singe housed and a cotton fiber nestlet (5.5 × 5.5 × 0.3 cm, ~ 2 g) was added to the cage. Mice were allowed 16 h to shred the nestlet and build the nest. Nest building was blindly scored, based on a 5-point rating scale: 1, nestlet not notably touched; 2, nestlet slightly torn; 3, nestlet mostly shredded, but no visible nest; 4, identifiable but flat nest; 5, fine nest. The average score was calculated and compared between genotypes.

## Delay and trace eyeblink conditioning

Stereotactic surgery was performed to implant a custom-built stainless headplate, as described previously[112]. Briefly, mice under isoflurane anesthesia were maintained on a stereotactic frame. An incision was made in the midline to expose the skull, and the overlying fascia was removed. The custom-built head plate was secured on the skull using screws and C&B Metabond cement (Parkell). Following the installation, the incision was closed with surgical glue and stitches. Mice were allowed to recover from surgical procedures for 7 d. Mice were acclimated to being head-restrained and running on a foam cylinder for 2 d. Mice were then given 1 training session per day. During the entire experiment, mice underwent a continuous training phase including 12 acquisition sessions, 3 extinction sessions, and 4 reacquisition sessions, and 2 wks later received 2 retention sessions. Each session consisted of 110 trials; the inter-trial interval was set to a random duration of 10−15 s. In all sessions during the acquisition, reacquisition, and retention, the CS paired with the US was delivered in each trial, except for every 10[th] trial in which only the CS was given. In those sessions during the extinction, the CS was presented without the US for all trials. For the delay eyeblink conditioning, each CS-US pairing consisted of a 350-ms CS of blue LED light (M470F1, Thorlabs), and a 30-ms US of air puff via a 23-gauge needle (6−8 psi), which was co-terminated with the CS. For the trace eyeblink conditioning, each CS-US pairing consisted of a 50-ms CS, a 270-ms stimulus-free interval, and a 30-ms US.

The training procedure was performed under infrared illumination by an LED light (IR56-56, CMVision), and eyelid position was captured by a high-speed (250 frames/s) monochrome camera equipped with a 25-mm lens (Mako G-040, Allied Vision). Subframes that contained only the eye and surrounding fur were acquired by image acquisition toolbox (MATLAB) interfaced with custom-written scripts. Off-line analysis of the individual frames was performed by custom-written MATLAB codes. Briefly, each frame was converted to binary by setting a threshold such that the value of each pixel of the eye is 0 and the value of each pixel of the fur is 1. The values of all the pixels were summed and the difference values were calculated by subtracting the average value of the baseline frames (100-ms duration, before CS-US pairing) from the value of individual frame throughout the trial. The ratio of the difference value to the average baseline value was further calculated and fraction eyelid closure (FEC) was obtained by normalizing these ratios into 0 (eye fully open) and 1 (eye fully closed). CRs were considered obtained if the FEC during the CS but prior to the US presentation is > 0.1. The percent CR in each trial was calculated by dividing the number of CRs by the total number of trials. CR amplitude was obtained by the peak FEC from every 10th trial, where only CS was given.

## Statistics

All behavioral experiments were performed blinded to the genotypes and treatment groups. The implementation of computer-assisted behavioral assessments and MATLAB scripts to analyze data provides additional assurance for unbiased data acquisition and analyses. All data were analyzed using Prism (GraphPad) and MATLAB. Comparisons between two groups were analyzed by two-tailed unpaired or paired Student's $t$ test or non-parametric Mann–Whitney test. Comparisons among more than two groups were analyzed by one-way ANOVA with Bonferroni's or Tukey's multiple comparison test or non-parametric Kruskal-Wallis test with Dunn's multiple comparison test. Two-way ANOVA repeated measures were used for analyses of the eyeblink conditioning in control and *Drd1* cKO groups. Sample sizes including the total number of experiments and the number of animals used are provided in the main text, or within associated main or supplemental figure legends. All data are shown as the mean ± SEM. Statistical differences were considered as significant at *$p < 0.05$, **$p < 0.01$, and ***$p < 0.001$.

## Reporting summary

Further information on research design is available in the Nature Portfolio Reporting Summary linked to this article.

## Data availability

Source data are provided with this paper. All data are available upon request to the Source data are provided with this paper.

## Code availability

The code for eyeblink conditioning video frame acquisition, processing, and analysis is available at GitHub[113] (https://zenodo.org/record/7569717#.Y_4sM-zMJ25). All relevant code is available upon request to the corresponding authors.

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

## Acknowledgements

We are grateful to Dr. Lucas Pozzo-Miller for the discussions and comments on the manuscript. We thank Yijian Zhang for maintaining mouse colony and writing MATLAB scripts for eyeblink conditioning. We thank Dr. Xin Xu and Cesar Acevedo for the discussions and Alejandro Osorio (University of Lausanne, Switzerland) for writing the initial MATLAB scripts. Funding was provided by grants from the United States National Institutes of Health (R21NS097913, R21NS108508, R21NS120315, and R01NS121542 to W.L.).

## Author contributions

W.L. designed the research. C.L., N.B.S., H.M., N.A.L., K.K., R.S., D.M., and W.L. conducted the research, analyzed the results. C.L., N.B.S., H.M., N.A.L., D.M., and W.L. wrote the manuscript.

## Competing interests

The authors declare no competing interests.
