## [Peer Review File · Nature Communications]

Purkinje cell dopaminergic inputs to astrocytes regulate cerebellar-dependent behaviorREVIEWER COMMENTS

Reviewer #1 (Remarks to the Author):

This study addresses the interesting and long-debated question concerning the possible sources of dopaminergic neuromodulation in the cerebellum. Two distinct hypotheses are proposed in the literature at present, one indicating a sparse and fragmented dopaminergic innervation of the cerebellar cortex from fibers originating in the VTA/SNc (Schweighofer et al., 2004), the other indicating Purkinje neurons (PN) as local sources of dopamine (Kim et al., 2009).

This report supports the latter view showing that 1) PN express one of the enzymes implicated in dopamine synthesis, 2) Bergmann glial cells (BG) express D1 receptors that elicit membrane depolarisations and cytosolic Ca²⁺ increases, and whose activation modulates excitatory and inhibitory synaptic transmission onto the PN.

The results shown are novel and original and the hypothesis of a loop involving DA release from PN, BG activation and glial modulation of PN synaptic inputs is appealing. However, although the results bring new insights on dopaminergic signalling in the cerebellar cortex, it is at present difficult to draw firm conclusions on the mechanisms at play from the data illustrated in this study. I have four major sources of concern:

1) It is not clear how PN may release dopamine and no specific mechanisms concerning this point are proposed in the paper. The authors demonstrate that PN express CYP2D that can metabolise tyramine into dopamine, but this is not sufficient to prove that they also release the neuromodulator. The authors show that electrical stimulation in slices elicits a release of dopamine in the extracellular space. Nevertheless, this stimulation is not specific to PN, and it may activate also other neuronal types, axonal afferents and glial cells. In this respect, the position of the stimulating electrodes in the cerebellar cortex should also be specified more precisely. More selective approaches may have been chosen to activate PN, for example chemo/optogenetic tools specifically targeting these cells (namely, excitatory DREADD receptors or ChR2 under the control of either the CAMKII or the L7 promoter, specific for PN).

2) The authors suggest that BG are the only cell type expressing D1 receptors (D1Rs) in the cerebellar cortex. To demonstrate this, they primarily rely on the DrD1a-Tomato mouse line. However, images in Figure 1 are over-exposed and Figure 1b shows some labelling in molecular layer processes that are GFAP-negative suggesting that D1Rs may be expressed in other cell types. Similarly, in situ hybridization seems to suggest the presence of D1 mRNA in the granular layer. Moreover, in order to prove the specificity of D1R labelling in BG processes, I believe it would be much more convincing to examine immunohistochemically the possible co-localization of D1Rs and GFAP (or S100 β) instead of using DrD1a-Tomato as a 'specific' BG marker (Figure 1d, e). Finally, a curiosity arises by looking at figure 1: are D1 receptors expressed in granular layer astrocytes?

3) In the brain, membrane depolarisations and Ca²⁺ mobilization from internal stores are typically associated with Gq-coupled receptor activation. In this study, in contrast, such effects are obtained in BG following activation of both the Gs-coupled D1Rs (shown to activate PKA via Gs in Figure 4f) and of hM3DGq DREADD receptors. Do the authors have an explanation for this apparent and counter-intuitive redundancy of signalling pathways in BG?

Furthermore, I have doubts on the relevance of the experiments shown in Figure 3b, the Ca²⁺

transients induced by depolarising pulses (Figure 3b) and those evoked by D1R activation (Figure 3c) not being comparable. Moreover: are Ca²⁺ responses induced by membrane depolarisations a “classical” response in BG? What are the molecular mechanisms that trigger these Ca²⁺ transients in purely passive glial cells?

4) Figure 5 is extremely confusing. The logical sequence of the experiments shown is hard to grasp. The effects of D1R activation on the excitatory and inhibitory synaptic inputs to PN should be studied by pharmacologically isolating the component under scrutiny, because the explanation proposed for the observed results (namely, Fig. 5b & d) is convoluted and unclear. As a notable example, the absence of BG from the scheme of Fig 5a is striking. Moreover, the sample traces of Fig. 5e illustrating the effects of D1R activation on PN EPSCs and IPSCs are not convincing. Finally, switching back and forth from BG activation via DR1 and chemogenetic methods is confusing. Overall, I thus think that the authors should show pharmacologically cleaner experiments and reorganize the narrative of the figure to make it more understandable and scientifically solid.

Minor points:

- p.6: is fluoxetine (I'm citing) a 'selective CYP2D inhibitor'?
- Figure 3b: what is the temporal window that has been fixed to study Ca²⁺ transients following BG depolarisations? What is the control condition?
- p. 7: I think it is 'gfaABC1D-GCaMP6f' and not 'ABC1D-GCaMP6f'.
- p. 8: the fact that BG express AMPA receptors lacking GluA2 subunit is well established by previous articles (Jonas & Burnashev 1995; Iino et al., 2001; Douyard et al., 2007; Saab et al., 2012). Figure 4a does not prove that GluA2 'was only present' in PNs because only PNs are labeled by calbindin.
- p.9 what are the 'n' of BG tested for AMPA receptor-dependent current rectification? And for NASPM experiments? Is there any statistics on this set of data?
- p10. In the Drd1 cKO, it would be better to quantify the biotinylated surface protein instead of Gria1 et Gria4 mRNA.
- Legend of figure 4b: in the middle of the panel there is a PN and not a BG as stated in the legend.
- Figure 5b: scale bars are missing.
- p.15 line 3: I think that is (Figure 7a, b) rather than (supplementary Fig.7a, b).
- p22: why do the authors use CNQX to block AMPA/Kainate receptors? It has been shown that this pharmacological agent has mixed inhibitory/excitatory response in cerebellar cortex neurons (Menuz et al. 2007).
- The paper from Nedergaard group (Wang, Xu, Wang.. et al., 2012) on BG-PN interaction must be discussed in this article.

Reviewer #2 (Remarks to the Author):

The authors have used a combination of immunocytochemistry, patch clamp recording, optical dopamine sensors and behavioral experiments in mutant mice to investigate dopamine signaling in the cerebellum.

They propose a model in which dopamine is synthesized in Purkinje cells though the enzymatic action of CYP2D and then released by unknown mechanisms to bind D1 receptors located on Bergmann glial cells. Action of the D1 receptor, through production of cAMP and activation of PKA

gives rise to depolarization and increased spontaneous Ca transients within the BG cells. These events, in turn, trigger yet-to-be defined processes that signal back to Purkinje cells, modulating their glutamatergic and GABAergic synaptic inputs. Interfering with the pathway by deleting D1 receptors in Bergmann glia in mutant mice results in impairments in social activity, eyeblink conditioning and locomotor activity.

Dopamine signaling in the cerebellum is an interesting topic with potential translational relevance and the authors have provided some intriguing and novel observations of high quality. Unfortunately, at present, there are many holes in the story and I remain unconvinced about both the details of the signaling pathway model and its relevance to the behaviors examined.

- 1). The immunostaining in Figure 2 showing CYP2D in Purkinje cells, which is the cornerstone of the model, must be validated by a control with tissue from the existing CYP2D null mouse.
- 2). The staining of AAV-CAG-dlight1.1 in Figure 2 shows several cell types labelled, not just Bergmann glia. It then becomes problematic to attribute the dlight signal to Bergmann glia specifically.
- 3). In Figure 2DE, the dlight transient is evoked by extracellular electrode stimulation of the cerebellar cortex so it then becomes impossible to know if the dopamine originated from the Purkinje cells or some other cellular compartment. Intracellular Purkinje cell stimulation would be illuminating here.
- 4). Attributing the action of fluoxetine on dlight signals to CYP2D inhibition is risky when this is the only manipulation used targeting CYP2D and fluoxetine has other known pharmacological effects. The existing CYP2D null mouse would be useful here.
- 5). If both SKF or chemogenetic activation of Bergmann glia are acting on Bergmann glial Ca transients downstream of depolarization then it should be possible to block these effects by hyperpolarizing the Bergmann glia cell with negative current injection.
- 6). In Figure 5D, the climbing fiber-evoked EPSCs seem to have an amplitude of about 300pA when $V_{hold} = -80$ mV (and there is no scale bar for Figure 5B). In my own experience, all-or-none CF-EPSCs in Cs-loaded Purkinje cells in this condition are at least 10-fold larger, to the point where they escape the voltage-clamp. I would need to see an I/O curve showing all-or-none responses to convince me that these are indeed CF-EPSCs.
- 7). The cre-lox strategy purported to selectively block the proposed cerebellar dopamine signaling pathway involves deleting the D1 receptor in cells expressing the glutamate transporter GLAST. While this will include Bergmann glial D1Rs, it will, of course, delete them in all cells of the body that express GLAST, including astrocytes throughout the central nervous system. It then becomes problematic to attribute the behavioral phenotypes specifically to actions at Bergmann glia.
- 8). I'm also left a bit confused about how the authors imagine this cerebellar dopamine signaling to operate in vivo. Dopamine release is presumably evoked by activating cerebellar Purkinje cells. Purkinje cells in vivo have a resting firing frequency of 40-100 Hz. Are we then to imagine that dopamine release from Purkinje cells is tonic, possibly modulated by firing frequency? Or do the authors see it as a phasic signal, perhaps only evoked by large Ca transients in the Purkinje cell as

produced during a climbing fiber-evoked complex spike? The experiments to explore these key points are straightforward and use techniques that the authors have already mastered.

9). It's unsatisfying that the authors haven't tested the hypothesis that dopamine release from Purkinje cells is vesicular (as opposed to resulting from reverse transporter action). Applying vesicular monoamine transporter inhibitor drugs to Purkinje cells through the patch pipette while measuring flight response to Purkinje cell activation would nail this point down.

Reviewer #3 (Remarks to the Author):

Summary:

The premise of the paper "Dopaminergic activity mediated by cerebellar astrocytes impacts Purkinje cells and cerebellum-related motor and social behavior" is of interest given that (1) D1 dopamine receptors are shown to be exclusively expressed in Bergmann glial cells of the cerebellum, (2) evidence is presented that dopamine is locally produced in Purkinje cells and binds to D1 receptors on Bergmann glial cells, and (3) some evidence is provided that following D1 receptor activation in Bergmann glial cells Purkinje cell excitability is reduced and (4) Selective knockout of D1 receptors in Bergmann glial cells are associated with impairments in cerebellum-dependent behaviors. Little is learned about under which physiological conditions Purkinje cells actually release dopamine to exert an effect -via Bergman glial cells- on the cerebellar network. Moreover, the exact mechanisms by which BGs exert their effect on excitability in the cerebellar network following D1 receptor activation remain unresolved. Even so, the authors provide new insights into dopaminergic signaling in the cerebellum through Bergmann glial cells. I commend the effort put into this study, but am left with a number of questions and doubts about the interpretability of some of the presented data.

Title:

The title of the manuscript is confusing as it does not reflect the essence of the findings accurately. It would probably be beneficial to change the title by being more specific about the astrocytes involved, i.e. replace cerebellar astrocytes with Bergmann glial cells. Change "cerebellum-related motor and social behavior" to "cerebellum-dependent behavior". Most importantly, the title does not provide information on the source of the Dopamine release, which the authors show are likely to be cerebellar Purkinje cells.

Abstract:

- "Astrocytic activities" is a broad, nondescript term revealing that the authors have not pinpointed exactly through which mechanism BGs' elevated activity affects excitability in cerebellar neurons.

Introduction:

- It seems as if the authors are confounding midbrain (p 2) and brain stem (p3)?
- (p 3) Authors could include recent reference of cerebellar involvement via VTA to food intake (2021 Low et al., Nature)
- Bergmann glial cells are implicated in neuropathological mechanisms of many diseases, but only one reference is provided alluding to involvement in autism.

Results:

- What I found missing in the text is the presentation of quantified results and statistics. Most information could be gleaned from the figure legends, but there were quite a few qualitative

statements in the main text that were not directly backed up by numbers. The authors should remedy this. For example, when the authors state that Purkinje cell spiking is significantly reduced upon D1R activation in BGs, it is unclear by how much the spike rate (where the authors don't seem to distinguish simple spikes from complex spikes!) was modulated. In some cases, observational data is hard to quantify, e.g. the presence of labeled projections from VTA. However, the figure panels are small and make it difficult to judge whether indeed there is no innervation, or sparse innervation from extracerebellar sources. The authors acknowledge that others have found sparse expression of extracerebellar dopaminergic fibers in the cerebellar cortex. Perhaps included larger panels in supplemental figures to show absence of sparse labeling (or provide raw data as source material).

- Although the immunohistochemistry in combination with smFish data convincingly show preferential expression of D1R in Bergmann glial cells, EM level data could have further driven home the point and localized the receptors to subcellular BG compartments.

- D1R expression is not confined to BGs in the cerebellum, but also present in the cerebellar nucleus (<https://www.sciencedirect.com/science/article/pii/S0006322318300672>). Homogenates of cerebellar slices were used to look at AMPAR surface expression following perfusion of a D1R agonist in the whole slice. Could expression of D1R in the cerebellar nuclei have affected the interpretation of their results?

- The term CB is introduced without defining that CB refers to Calbindin.

- The CYP2D inhibitor fluoxetine may be specific to block CYP2D and prove the origin of dopamine release from Purkinje cells that express it, but it should be noted that fluoxetine is not a clean drug, and can increase GABA currents (J Pharmacol Exp Ther, 2003, doi: 10.1124/jpet.102.044834). In this context, I thought the statement that fluoxetine being selective was a bit misleading.

- The authors show that D1 receptor activation depolarizes BGs and increases the frequency of calcium signals and their peak amplitude in these cells. A supplemental movie is provided that shows supposedly increased frequency of calcium signals after D1R activation, but there is no baseline. I would therefore include two movies one showing baseline and one showing calcium signals following D1R activation.

- The authors show that D1R activation in BGs results increases the frequency of calcium signals in BGs via release from internal stores. They also show that it can increase surface expression of Ca²⁺-permeable AMPA receptors that are associated with an increase in calcium levels in BGs. It would probably be good if the authors add a statement about the physiological impact of these two sources of calcium in BGs. For example, the ensheathment of synapses by BG processes depends on AMPA-R mediated calcium influx. Release of calcium from internal stores could trigger other processes that affect signaling from BGs to neurons. These two sources of calcium influx are not disambiguated in the manuscript. Also make clear which conclusions you are unable to draw from your data, rather than leave it open-ended.

- The authors acknowledge previous studies that have shown the presence of slow depolarizing currents driven by activation of D2R on Purkinje cells, suggesting an autocrine function where dopamine that is released from PCs acts on the same cells. If dopamine is released from PCs and binds to surface expressed dopamine receptors in BGs, it could also bind to dopamine receptors on PCs. How does this affect interpretation of the author's results?

- The authors suggest that (based on cFos immunostaining in combination with a conditional

knockout of D1R in BGs of the cerebellum not showing any impairments in behavior,) the cerebellum is not involved in delayed eyeblink conditioning. Even if this is what the authors find, it is a very contentious statement that is not backed up by most other studies. In fact, there is overwhelming empirical support for the involvement of the cerebellum in delayed eyeblink conditioning. The authors suggest that the interval used for conditioning may have deviated from other studies, but it is identical to the interval used in studies that clearly show cerebellar involvement. Moreover, several cells in the cerebellum are intrinsically active (Purkinje cells, cerebellar nuclear neurons to name a few). This will possibly impact the ability to make conclusive statements about neuronal activity in the cerebellum using cFos staining.

- The authors show that MLIs are activated after D1R activation in BGs, but the exact mechanism through which this happens remains unclear. This should be acknowledged.
- The authors state: "It should be noted that in a few cases especially in CF-PC synapses, there existed a transient increase in PSC amplitude lasting about 1-2 min." We don't see this data reported. It would be good to include the data as a supplemental figure.
- The authors should more clearly emphasize throughout the text that the cKO is specific to BG in the cerebellar cortex (bearing in mind that astrocytes in other regions may potentially also be affected).
- In the transgenic animals use D1 receptors are selectively knocked out in BGs, but what do the authors believe is the likelihood that D1 receptor function in other astrocytes are also affected (e.g. <https://www.frontiersin.org/articles/10.3389/fnana.2017.00003/full>) and thus contribute to the reported impairments in learning and behavior? Some evidence of selectivity is presented by infusing D1R agonist directly into the cerebellum of WT mice (which may not have enough specificity to avoid also activating CN neurons expressing D1 receptors). However, this is only shown for the open field test.
- Figure 5g shows a decline in firing frequency, but also change in the regularity of firing. Can the authors quantify firing regularity (e.g. CV2)?
- The authors mention achieving "near-super resolution" when studying localization of GluA1, GluA4, but do not provide figures for the exact resolution obtained (neither found in the main text, nor methods section). Please add this information.
- This manuscript really calls for a schematic to demonstrate the (putative) modes of action of locally released DA in the cerebellar circuit.

Figures:

- In Figure 2 the box used to calculate background fluorescence can hardly be seen. Change color for clarity.
- Figure 3 panels (a) and (b) should show colors matching to those in other bar graphs
- Supplemental Figure 5, please indicate which colors represent WT, or cKO.

We thank three reviewers for their informed and constructive comments, which greatly improve our manuscript. We have responded to the comments as detailed below. The changes made to the manuscript are highlighted in yellow.

Reviewer #1

Major Points:

1. *It is not clear how PN may release dopamine and no specific mechanisms concerning this point are proposed in the paper. The authors demonstrate that PN express CYP2D that can metabolise tyramine into dopamine, but this is not sufficient to prove that they also release the neuromodulator. The authors show that electrical stimulation in slices elicits a release of dopamine in the extracellular space. Nevertheless, this stimulation is not specific to PN, and it may activate also other neuronal types, axonal afferents and glial cells. In this respect, the position of the stimulating electrodes in the cerebellar cortex should also be specified more precisely. More selective approaches may have been chosen to activate PN, for example chemo/optogenetic tools specifically targeting these cells (namely, excitatory DREADD receptors or ChR2 under the control of either the CAMKII or the L7 promoter, specific for PN).*

Response: We thank the reviewer for pointing out this important issue, regarding the DA release. In our new set of experiments (**Fig. 2f, g**; also see the text on page 8), to demonstrate whether DA is released directly from PCs, we injected both AAV-DIO-hM3D(Gq)-mCherry and AVV-dLight1.1-GFP into the cerebellum of *Pcp2-Cre* mice for expression of excitatory DREADD in PCs and dLight1.1 primarily in BGs (**Supplementary Fig. 4a**), respectively. We first confirmed that hM3D(Gq) and dLight1.1 could be expressed in close proximity to each other in the cerebellum and that CNO activation could induce PC depolarization and firing. We then imaged the slices during CNO perfusion and obtained dLight1.1 signals by subtracting the background signals. We found that CNO treatment significantly induced dLight1.1 signals. This chemogenetic approach, as suggested by the reviewer, provides strong evidence that in addition to DA synthesis, PC can directly release it.

2. *The authors suggest that BG are the only cell type expressing D1 receptors (D1Rs) in the cerebellar cortex. To demonstrate this, they primarily rely on the DrD1a-Tomato mouse line. However, images in Figure 1 are over-exposed and Figure 1b shows some labelling in molecular layer processes that are GFAP-negative suggesting that D1Rs may be expressed in other cell types. Similarly, in situ hybridization seems to suggest the presence of D1 mRNA in the granular layer. Moreover, in order to prove the specificity of D1R labelling in BG processes, I believe it would be much more convincing to examine immunohistochemically the possible co-localization of D1Rs and GFAP (or S100 β) instead of using DrD1a-Tomato as a 'specific' BG marker (Figure 1d, e). Finally, a curiosity arises by looking at figure 1: are D1 receptors expressed in granular layer astrocytes?*

Response: The purpose of overexposing the image in **Fig. 1a** is to demonstrate the clear expression of D1R in the cerebellum. D1R expression in the cerebellum, as expected, is lower than the striatum in *Drd1a*-tdTomato mice; however, it is explicitly higher than any other posterior brain regions and can be clearly observed.

Immunolabeling of intermediate filaments with anti-GFAP was not able to reveal the entire BG morphology (**Fig. 1b**), leaving other fine processes in the ML unlabeled. To confirm that D1Rs are primarily enriched in BGs but not in other cell types, we performed additional experiments. First, we performed co-immunolabelling of S100 β and tdTomato in the cerebellum of *Drd1a*-tdTomato mice. As S100 β is a calcium-binding protein, which is distributed in the cytoplasm and nucleus, co-immunolabelling with their two antibodies exhibits better colocalization across the nearly entire BGs (**Supplementary Fig. 1b**). Secondly, we filled BGs from *Drd1a*-tdTomato mice with biocytin followed by incubation with streptavidin-conjugated Alexa Fluor 488. Again, all those processes that were unlabeled with anti-GFAP were stained by infused biocytin (**Fig. 1c**). In addition to using *Drd1a*-tdTomato reporter mice, we corroborate the D1R expression in BGs using the selective D1R antibody, as suggested by the reviewer. Our original manuscript demonstrates that D1Rs detected by the antibody are distributed in *Drd1a*-tdTomato-labeled BGs. Since biocytin can penetrate throughout the entire BGs, we performed D1R labeling using the same antibody in combination with biocytin infusion. Our data show that D1Rs can be found in biocytin-filled BGs (**Fig. 1e**), consistent with other results.

We also thank the reviewer for pointing out the possible expression of D1Rs in cell types in the GCL based on our smFISH data. To determine this possibility, we imaged this layer in *Drd1a*-tdTomato mice, which was co-immunostained with NeuN or GFAP and tdTomato. We did not find the D1R expression in either NeuN-positive GCs or GFAP-positive astrocytes in the GCL (**Supplementary Fig. 2a**).

- In the brain, membrane depolarisations and Ca²⁺ mobilization from internal stores are typically associated with Gq-coupled receptor activation. In this study, in contrast, such effects are obtained in BG following activation of both the Gs-coupled D1Rs (shown to activate PKA via Gs in Figure 4f) and of hM₃DGq DREADD receptors. Do the authors have an explanation for this apparent and counter-intuitive redundance of signalling pathways in BG?*

Furthermore, I have doubts on the relevance of the experiments shown in Figure 3b, the Ca²⁺ transients induced by depolarising pulses (Figure 3b) and those evoked by D1R activation (Figure 3c) not being comparable. Moreover: are Ca²⁺ responses induced by membrane depolarisations a “classical” response in BG? What are the molecular mechanisms that trigger these Ca²⁺ transients in purely passive glial cells?

Response: We appreciate the reviewer for this insightful comment on the type of G proteins activated by DA in the cerebellum. Our data show that the D1R agonist SKF83822 promotes GluA1 insertion, which can be blocked by protein kinase A (PKA) inhibitor Rp-cAMPs. As a result, the number of GluA2-lacking Ca²⁺-permeable (CP)-AMPA receptors are increased in BG membrane. Ca²⁺ entry is expected to be enhanced. This result suggests the activation of the Gs member by D1R activation. Furthermore, we also showed that D1R activation by SKF83822 induces Ca²⁺ transients, which can be blocked by the IP₃R antagonist 2-APB. This finding can be mimicked by a chemogenetic approach involving excitatory hM₃DGq DREADD receptors. This finding suggests the activation of Gq member by D1R activation. Collectively, our data suggest both Gs and Gq signaling can be initiated by D1R action in the cerebellum. The classical view holds that D1R signaling can be only coupled with Gs, but accumulating evidence from many

studies suggests that D1R can stimulate Gq (see Lee SP, *J Biol Chem* 279, 2004; Banday AA, *American Journal of Physiology-Renal Physiology* 293, 2007; Liu J, et al. *Neuropharmacology* 57, 2009; Mizuno K, *J Neurochem* 122, 2012; Undie, *Eur J Pharmacol* 226, 1992; Yu Y, *J Neurochem* 104, 2008; Medvedev, *J Neurosci* 33, 2013). Consistent with our results, a study showed that 2-APB treatment could block D1R-induced the PLC-IP3R signaling (See Fieblinger T, *J Neurosci* 34, 2014). More notably, there is also evidence demonstrating that D1R also stimulates Gq in astrocytes (see Liu L, *Cell Mol Neurobiol* 29, 2009; Zhang X, *J Neurosci* 29, 2009). Together with these previous reports, we believe that D1R activation can stimulate both Gs and Gq, which may function distinguishingly or maybe interactively in triggering Ca²⁺ rises in BGs. We have incorporated this discussion into the manuscript (pages 19-20).

We have removed from the main text the data concerning Ca²⁺ transients induced by depolarizing pulses. Considering the Ca²⁺ transients activated by D1Rs may be partially caused by its depolarizing effect, we kept it as the supplemental data (**Supplementary Fig. 5a**). Ca²⁺ transients in BGs are caused by many factors including ATP, K⁺, kainate, and mGluR5 activators (see Verkhratsky A, *Function* 1, 2020). Although membrane depolarization is not the only factor contributing to Ca²⁺ elevations, it is an important cellular process in BGs, which is shared by neurons during D1R activation.

4. *Figure 5 is extremely confusing. The logical sequence of the experiments shown is hard to grasp. The effects of D1R activation on the excitatory and inhibitory synaptic inputs to PN should be studied by pharmacologically isolating the component under scrutiny, because the explanation proposed for the observed results (namely, Fig. 5b & d) is convoluted and unclear. As a notable example, the absence of BG from the scheme of Fig 5a is striking. Moreover, the sample traces of Fig. 5e illustrating the effects of D1R activation on PN EPSCs and IPSCs are not convincing. Finally, switching back and forth from BG activation via DR1 and chemogenetic methods is confusing. Overall, I thus think that the authors should show pharmacologically cleaner experiments and reorganize the narrative of the figure to make it more understandable and scientifically solid.*

Response: We appreciate the comment that the reviewer gave on this figure. We have made substantial modifications to this figure. The finding of the amplitude reduction of PSC (without blockade of GABAergic activity) during the D1R activation by SKF83822 is very surprising (**Fig. 5b**), which prompted us to speculate that PSCs in our recordings contain two mixed components: EPSCs and IPSCs. The IPSC component was confirmed by recordings in PCs held at different voltages (**Supplementary Fig. 7e**). The enhancing effect of D1R activation on IPSCs was also verified (**Supplementary Fig. 7f**). Based on these confirmations, we believe that the data in **Fig. 5b** are good to show. Experiments designed to examine pharmacologically isolated evoked EPSCs and IPSCs are great, but we are confident that examining the isolated spontaneous EPSCs and IPSCs (**Fig. 5c**) is adequate to answer the question concerning the important role of D1R action on excitatory and inhibitory synaptic transmission.

We have added BGs to the schematic drawing (**Fig. 5a**). We have added new recording traces to **Fig. 5c** to better illustrate the effect of D1R activation on sEPSCs and sIPSCs. To make this figure clear, we have removed all experiments with the use of chemogenetic stimulation. As the data in **Fig. 3** show the similar effect of chemogenetic stimulation as the pharmacological D1R activation, a logic question that readers would

ask is whether it has the similar effect on PC synaptic transmission. Thus, we retain it as supplemental data (**Supplementary Fig. 7h**). We hope that reorganizing this figure and replacing example traces make this figure more succinct and comprehensible.

Minor Points:

5. *p.6: is fluoxetine (I'm citing) a 'selective CYP2D inhibitor'?*

Response: We have replaced 'selective CYP2D inhibitor' with 'potent CYP2D inhibitor' (page 7).

6. *Figure 3b: what is the temporal window that has been fixed to study Ca²⁺ transients following BG depolarisations? What is the control condition?*

Response: We recorded Ca²⁺ signals for a total of 10 min. Ca²⁺ frequency or peak $\Delta F/F$ was compared before (2-min duration) with after current injections (8-min duration). Control BGs were held at -80 mV during Ca²⁺ imaging with Fluo 5F. I have added the description to the text (pages 26-27).

7. *p. 7: I think it is 'gfaABC1D-GCaMP6f' and not 'ABC1D-GCaMP6f'.*

Response: Corrected (page 9).

8. *p. 8: the fact that BG express AMPA receptors lacking GluA2 subunit is well established by previous articles (Jonas & Burnashev 1995; Iino et al., 2001; Douyard et al., 2007; Saab et al., 2012). Figure 4a does not prove that GluA2 'was only present' in PNs because only PNs are labeled by calbindin.*

Response: We thank the reviewer for bringing up these references. Our thorough examination of these AMPAR subunits in BGs agrees with these previous findings. We have cited these articles (page 11). To further prove that GluA2 is present in PCs but not in BGs, we co-immunostained GluA2 with the BG marker GFAP in control mice (**Supplementary Fig. 6a**) and tdTomato in *Drd1a*-tdTomato mice (**Supplementary Fig. 6b**). We found no colocalization. The observation that GluA2 is absent in BGs is clear, but whether GluA2 is present in other various types of interneurons is not determined. To make this statement unambiguous, we have removed "only" from the text (page 11).

9. *p.9 what are the 'n' of BG tested for AMPA receptor-dependent current rectification? And for NASPM experiments? Is there any statistics on this set of data?*

Response: We have added 'n' to inward rectification (**Fig. 4c**) and NASPM (**Supplementary Fig. 5c**) experiments. Since the electrophysiological and Ca²⁺ signal features are expected for GluA2-lacking AMPARs, we did not perform statistical analyses on these data but presented it as a demonstration.

10. *p.10. In the *Drd1* cKO, it would be better to quantify the biotinylated surface protein instead of *Gria1* et *Gria4* mRNA.*

Response: We have performed additional experiments on surface GluA1, GluA2, and GluA2 expression in *Drd1* cKO, which replaced the quantitative RT-PCR data (**Fig. 4g**).

Our data show that the surface levels of GluA1 and GluA4 but not GluA2 are downregulated.

11. *Legend of figure 4b: in the middle of the panel there is a PN and not a BG as stated in the legend.*

Response: We have added “PC” to the sentence (page 63).

12. *Figure 5b: scale bars are missing.*

Response: We have added the scale bar to this panel (**Fig. 5b**).

13. *p.15 line 3: I think that is (Figure 7a, b) rather than (supplementary Fig.7a, b).*

Response: Corrected (page 17).

14. *p22: why do the authors use CNQX to block AMPA/Kainate receptors? It has been shown that this pharmacological agent has mixed inhibitory/excitatory response in cerebellar cortex neurons (Menuz et al. 2007).*

Response: We thank the reviewer for raising this important issue. Indeed, this study demonstrates non-selectivity of CNQX in blocking AMPAR activity. In our experiments, CNQX was used to block membrane currents induced by AMPA puffing but not electrical stimulation. We believe that the effect from other receptor types or synaptic inputs should be minimal.

15. *The paper from Nedergaard group (Wang, Xu, Wang.. et al., 2012) on BG-PN interaction must be discussed in this article.*

Response: We have discussed this paper regarding BG-PC interaction in the manuscript (page 20).

Reviewer #2

1. *The immunostaining in Figure 2 showing CYP2D in Purkinje cells, which is the cornerstone of the model, must be validated by a control with tissue from the existing CYP2D null mouse.*

Response: We thank the reviewer for raising this fundamental issue. Our immunostaining data provide the evidence for the expression CYP2D in PCs of the cerebellum. Although there is a lack of references showing the role of DA in the cerebellum, there have been many reports showing the distribution of CYP2D in the cerebellum (see Cheng J. *Xenobiotica* 43, 2013; Miksys S. *Xenobiotica* 30, 2000; Miksys S. *Drug Metab Dispos* 33, 2005; Li J. *Br J Pharmacol* 172, 2015; Sheng Y. *Front Pharmacol* 12, 2021; Kuban W. *Drug Metabolism Reviews* 53, 2021), or more specifically in PCs (see Siegle I. *Pharmacogenetics* 11, 2001; Chinta SJ. *Brain Res Mol Brain Res* 103 2002; Miksys S. *J Neurochem* 82, 2002). We attempted to confirm these results in CYP2D knockout mice; unfortunately, these live mice are not commercially available at this moment (<https://www.jax.org/strain/002910>). We believe that together with those published data, CYP2D is highly expressed in PCs.

2. *The staining of AAV-CAG-dLight1.1 in Figure 2 shows several cells types labelled, not just Bergmann glia. It then becomes problematic to attribute the dLight signal to Bergmann glia specifically.*

Response: In our newly added experiments using immunostaining, we found that dLight1.1 is largely distributed in S100 β -expressing BGs but not in CB-expressing PCs (**Supplementary Fig. 4a**), suggesting that dLight1.1 is preferentially expressed in BGs. Indeed, if CAG-dLight1.1 is only expressed in BGs, this will make our model more explainable. However, the bottom line is that irrespective of the location of its expression, the increase of dLight1.1 signals suggests the release of DA in the extracellular milieu. D1Rs in BGs are expected to be activated by the released DA. If in the future, there are viruses available with the promoter driving dLight1.1 expression in BGs, it will be interesting to further examine DA release in a spatiotemporal manner.

3. *In Figure 2DE, the dLight transient is evoked by extracellular electrode stimulation of the cerebellar cortex so it then becomes impossible to know if the dopamine originated from the Purkinje cells or some other cellular compartment. Intracellular Purkinje cell stimulation would be illuminating here.*

Response: Indeed, field stimuli confound the determination of the origin of DA release. As suggested by the reviewer, we recorded dLight1.1 signals during PC firing triggered by positive current injections (**Supplementary Fig. 4d**; also see the text on page 8). However, we could not detect any change in dLight1.1 signals. We speculate that because PCs are not strongly electronically coupled or may be strongly inhibited by interneurons, action potentials from a single PC are insufficient to induce detectable DA release. We then resorted to the chemogenetic approach that can be used to induce firing of a large population of PCs. We were able to detect DA release by specifically targeting PCs. For more explanation on chemogenetic stimulation, we refer the reviewer to Major Point 1 in Reviewer 1.

4. *Attributing the action of fluoxetine on dLight signals to CYP2D inhibition is risky when this is the only manipulation used targeting CYP2D and fluoxetine has other known pharmacological effects. The existing CYP2D null mouse would be useful here.*

Response: To confirm the effect shown by fluoxetine treatment, in our additional experiments, we took advantage of another potent CYP2D inhibitor quinidine. Quinidine or quinine has been successfully used to block CYP2D function following its treatment of animal brain microsomes (see Bromek E. *European Journal of Pharmacology* 626, 2010) or its systematic administration to animals (see Bromek E. *European Journal of Pharmacology* 626, 2010). Following the treatment of brain slices with quinidine (**Supplementary Fig. 4b**), we also observed a significant reduction of dLight1.1. Based on these two sets of independent experiments, we strongly believe that DA in PCs is synthesized through CYP2D.

5. *If bath SKF or chemogenetic activation of Bergmann glia are acting on Bergmann glial Ca transients downstream of depolarization then it should be possible to block these effects by hyperpolarizing the Bergmann glia cell with negative current injection.*

Response: We thank the reviewer for this great suggestion. We have attempted to test

if membrane hyperpolarization thwarts SKF83822-induced Ca^{2+} signals. Interestingly, however, the hyperpolarization itself also increased Ca^{2+} activity (**Supplementary Fig. 5a**; also see the text on page 9). This hyperpolarization-induced Ca^{2+} influx may be caused by activation of I_h channels (see Yu X. *Proc Natl Acad Sci U S A* 101, 2004). Although SKF83822-induced Ca^{2+} rises are mechanistically different from hyperpolarization-induced Ca^{2+} entry, our current Ca^{2+} imaging technique is not capable of determining whether hyperpolarization inhibits depolarization-induced Ca^{2+} transients, considering the overall increase of Ca^{2+} signals in the cytoplasm.

6. *In Figure 5D, the climbing fiber-evoked EPSCs seem to have an amplitude of about 300pA when $V_{\text{hold}} = -80 \text{ mV}$ (and there is no scale bar for Figure 5B). In my own experience, all-or-none CF-EPSCs in Cs-loaded Purkinje cells in this condition are at least 10-fold larger, to the point where they escape the voltage-clamp. I would need to see an I/O curve showing all-or-none responses to convince me that these are indeed CF-EPSCs.*

Response: We appreciate the reviewer's comment on this technical issue. The absolute size of a single evoked EPSC at CF-PC synapses depends on many conditions, including intracellular solution composition, pipette and input resistances, and the distance between the stimulus and recording pipettes. We have added the scale bar to **Fig. 5b**. For CF-PC recordings, we normally set the stimulus intensity below the level where an unclamped recording in PCs is triggered. The size of this example EPSC is approximately 100 pA before the treatment. To confirm the proper placement of stimulus electrodes, we routinely build an I/O curve and examine its slope. New example recordings have been added to show the I/O curve form PF-PC and CF-PC synapses (**Supplementary Fig. 7a**). We show that with the increasing stimulus intensity graded EPSCs are evoked at PF-PC synapses whereas all-or-none EPSCs are evoked at CF-PC synapses.

7. *The cre-lox strategy purported to selectively block the proposed cerebellar dopamine signaling pathway involves deleting the D1 receptor in cells expressing the glutamate transporter GLAST. While this will include Bergmann glial D1Rs, it will, of course, delete them in all cells of the body that express GLAST, including astrocytes throughout the central nervous system. It then becomes problematic to attribute the behavioral phenotypes specifically to actions at Bergmann glia.*

Response: We thank the reviewer for raising this important issue. It is true that in our *Drd1* cKO mice, all cells expressing the GLAST are deficient in D1Rs. Since GLAST is an astrocytic glutamate transporter, D1Rs should be only deleted from astrocytes. In our complementary experiments, we examined the expression of D1R in astrocytes in several major brain regions by using coimmunostaining of GFAP or s100 β and tdTomato in *Drd1a*-tdTomato mice (**Supplementary Fig. 2c-f**). These brain regions include cerebellar cortex, hippocampus, and striatum, which are important for brain functions including motor and social behaviors. We did not observe any colocalization of tdTomato with either astrocyte marker, suggesting there is no expression of D1R in these astrocytes. Knockout of D1Rs from these astrocytes would be unlikely to result in a significant effect on animal behavior. Even though we cannot completely exclude the possibility of the behavioral effect of D1R deletion in astrocytes in other small brain

regions, we believe that the large population of BGs that are deficient in D1Rs should play a major role in motor and social deficits.

8. *I'm also left a bit confused about how the authors imagine this cerebellar dopamine signaling to operate in vivo. Dopamine release is presumably evoked by activating cerebellar Purkinje cells. Purkinje cells in vivo have a resting firing frequency of 40-100 Hz. Are we then to imagine that dopamine release from Purkinje cells is tonic, possibly modulated by firing frequency? Or do the authors see it as a phasic signal, perhaps only evoked by large Ca transients in the Purkinje cell as produced during a climbing fiber-evoked complex spike? The experiments to explore these key points are straightforward and use techniques that the authors have already mastered.*

Response: To test DA release in freely moving animals, we have performed *in vivo* fiber photometry in the cerebellum expressing dLight1.1. We observed spontaneous DA release in the cerebellum (**Fig. 2i**). We also compared DA release in the cerebellum with that in the striatum. We found that the spontaneous signals in the cerebellum were 3-4-fold less frequent than those in the striatum. In the nigrostriatal pathway, DA is released in the striatum in response to tonic (~4 Hz) and phasic (20-100Hz) patterns of neuronal firing in the VTA or SNc. In the cerebellum, PF activity-related simple spikes in PCs occur at 40-100Hz and CF activity-related complex spikes occur at 1Hz. The fold difference of the firing frequency between tonic activity in DAergic neurons and complex spikes in PCs is similar to that of the DA release between these two activities. Considering this fact with massive depolarization bursts generally induced by CFs, we attempt to speculate that DA release may be followed by complex spikes. However, we cannot exclude the possibility of DA release induced by PF-related PC firing. To determine these possibilities, our experiments in the future will monitor DA release in combination with *in vivo* single- or multi-unit recordings in the cerebellum and analyze the correlation between DA release and simple or complex spikes.

9. *It's unsatisfying that the authors haven't tested the hypothesis that dopamine release from Purkinje cells is vesicular (as opposed to resulting from reverse transporter action). Applying vesicular monoamine transporter inhibitor drugs to Purkinje cells through the patch pipette while measuring dlight response to Purkinje cell activation would nail this point down.*

Response: We thank the reviewer for his comment on the routes of DA release. As our experiments indicated the inability to detect DA release following a single PC activity (**Supplementary Fig. 4d**), we chose bath application of the vesicular monoamine transporter inhibitor reserpine instead of filling the cell with it. Our data showed that dLight1.1 signals were significantly decreased, indicating that DA release is associated with the action of vesicular monoamine transporters (**Fig. 2h**; also see the text on page 8).

Reviewer #3

Title:

1. *The title of the manuscript is confusing as it does not reflect the essence of the findings accurately. It would probably be beneficial to change the title by being more specific*

about the astrocytes involved, i.e. replace cerebellar astrocytes with Bergmann glial cells. Change "cerebellum-related motor and social behavior" to "cerebellum-dependent behavior". Most importantly, the title does not provide information on the source of the Dopamine release, which the authors show are likely to be cerebellar Purkinje cells.

Response: We thank the reviewer for the great suggestion. We have changed the title to “*Purkinje cell dopaminergic inputs to Bergmann glia regulate cerebellar-dependent behavior*”.

Abstract:

- 2. “Astrocytic activities” is a broad, nondescript term revealing that the authors have not pinpointed exactly through which mechanism BGs’ elevated activity affects excitability in cerebellar neurons.*

Response: We found BG depolarization and Ca²⁺ rises following D1R activation. We also showed that this activation results in an increased inhibitory but a decreased excitatory input to PCs. We believe that these actions are likely to through glutamate release from BGs, but at this point we do not have direct evidence to pinpoint this exact mechanism. We have discussed this aspect in more detail in the Discussion section (pages 20-21).

Introduction:

- 3. It seems as if the authors are confounding midbrain (p 2) and brain stem (p3)?*

Response: We appreciate the reviewer’s comment. We have replaced “the brainstem” with “the midbrain”.

- 4. (p 3) Authors could include recent reference of cerebellar involvement via VTA to food intake (2021 Low et al., Nature)*

Response: We have incorporated this recent paper into the Introduction section (page 3).

- 5. Bergmann glial cells are implicated in neuropathological mechanisms of many diseases, but only one reference is provided alluding to involvement in autism.*

Response: We have added two more articles that highlight the importance of BGs in neurological diseases (page 4).

Results:

- 6. What I found missing in the text is the presentation of quantified results and statistics. Most information could be gleaned from the figure legends, but there were quite a few qualitative statements in the main text that were not directly backed up by numbers. The authors should remedy this. For example, when the authors state that Purkinje cell spiking is significantly reduced upon D1R activation in BGs, it is unclear by how much the spike rate (where the authors don’t seem to distinguish simple spikes from complex spikes!) was modulated. In some cases, observational data is hard to quantify, e.g. the presence of labeled projections from VTA. However, the figure panels are small and make it difficult to judge whether indeed there is no innervation, or sparse innervation*

from extracerebellar sources. The authors acknowledge that others have found sparse expression of extracerebellar DAergic fibers in the cerebellar cortex. Perhaps included larger panels in supplemental figures to show absence of sparse labeling (or provide raw data as source material).

Response: We have thoroughly examined the manuscript and added the numbers and statistics. For those figures with graphs, we added all the missing numbers (**Fig. 4c**). For those example traces without statistical analysis, we added the numbers if the multiple independent experiments were performed (**Fig. 2c; Fig. 2d; Supplementary Fig. 4c; Supplementary Fig. 4d; Supplementary Fig. 6c; Supplementary Fig. 7f**). For those continuous recordings with the comparisons only made between pre- and posttreatment, the statistical methods and *p* values were added in the corresponding figure legends (**Fig. 5c; Fig. 5d; Supplementary Fig. 7h**).

We apologize for the error we made for the label of *y*-axis in **Fig. 5d**. It should be the normalized spike frequency. Furthermore, we added the actual values of the spike rate before and after treatment in the legend for the readers' reference. However, when we analyze these data, we compared the spike frequency before and after treatment for each individual cell and then calculated for the average fold change. The rate of complex spikes recorded in cell-attached mode is very low; our data also show D1R activation has a very similar effect on PF and CF input to PCs. Based on these facts, we did not distinguish the rate of simple spikes from complex spikes when we calculated the spike frequency.

To better demonstrate the minimal labelling of DA axonal afferents from the SNc/VTA or LC, we have provided each panel with schematic drawing and enlarged all these images for clear illustration (**Supplementary Fig. 3**).

7. *Although the immunohistochemistry in combination with smFISH data convincingly show preferential expression of D1R in Bergmann glial cells, EM level data could have further driven home the point and localized the receptors to subcellular BG compartments.*

Response: Our immunostaining and smFISH data indicate that D1Rs are located in BGs. We added two more experiments to demonstrate the expression of D1Rs in BGs. First, we filled BGs from *Drd1a*-tdTomato slices with biocytin followed by immunostaining with antibody against tdTomato (**Fig. 1c**). We found that the fine processes of BGs are also enriched with D1Rs. Second, we filled BGs from control mice with biocytin followed by immunostaining anti-D1R (**Fig. 1e**). Again, we observed the colocalization of BG processes with D1Rs. We agree that EM can provide more information regarding the subcellular localization. Because our work mainly focuses DA action on BGs, we believe that our present data have adequately answered this question. A future work in our lab will perform EM analysis and determine how DR1s interact with other receptors or molecules to induce the mobilization of many releasable factors in BGs.

8. *D1R expression is not confined to BGs in the cerebellum, but also present in the cerebellar nucleus
(<https://www.sciencedirect.com/science/article/pii/S0006322318300672>).
Homogenates of cerebellar slices were used to look at AMPAR surface expression*

following perfusion of a D1R agonist in the whole slice. Could expression of D1R in the cerebellar nuclei have affected the interpretation of their results?

Response: Our complementary data indeed showed that D1Rs are expressed in the DCN. They are mainly on neurons but not in astrocytes of the DCN (**Supplementary Fig. 2b**). We agree that homogenates of cerebellar slices also contain a fraction of D1Rs from neurons in DCN. Considering the larger number of BGs and the higher expression level of D1Rs in BGs, however, we believe that our data from the whole cerebellar slices should reflect the change in BGs.

9. *The term CB is introduced without defining that CB refers to Calbindin.*

Response: We have spelled out the full term for CB when introducing it for the first time in the text (page 5).

10. *The CYP2D inhibitor fluoxetine may be specific to block CYP2D and prove the origin of dopamine release from Purkinje cells that express it, but it should be noted that fluoxetine is not a clean drug, and can increase GABA currents (J Pharmacol Exp Ther, 2003, doi: 10.1124/jpet.102.044834). In this context, I thought the statement that fluoxetine being selective was a bit misleading.*

Response: We thank the reviewer for pointing out this issue. We have replaced “a selective CYP2D inhibitor fluoxetine” with “a potent CYP2D inhibitor fluoxetine”. To confirm the effect that was not derived from its side effect, we also tested another CYP2D inhibitor quinidine. We observed a significant reduction of dLight1.1 following the treatment (**Supplementary Fig. 4b**). For more information regarding the use of quinidine, we refer the reviewer to Point 4 in Reviewer 2.

11. *The authors show that D1 receptor activation depolarizes BGs and increases the frequency of calcium signals and their peak amplitude in these cells. A supplemental movie is provided that shows supposedly increased frequency of calcium signals after D1R activation, but there is no baseline. I would therefore include two movies one showing baseline and one showing calcium signals following D1R activation.*

Response: We appreciate the reviewer’s comment. Since the baseline we acquired for the movie lasts about 2 min, the first movie consists of both the baseline and the following SKF83822 treatment. To further match the figure, we have added another movie that was taken for the slice pretreated with 2-APB. This treatment reduced Ca²⁺ transients induced by SKF83822 to a level equivalent to the baseline (**Supplementary Movie 1**).

12. *The authors show that D1R activation in BGs results increases the frequency of calcium signals in BGs via release from internal stores. They also show that it can increase surface expression of Ca²⁺-permeable AMPA receptors that are associated with an increase in calcium levels in BGs. It would probably be good if the authors add a statement about the physiological impact of these two sources of calcium in BGs. For example, the ensheathment of synapses by BG processes depends on AMPA-R mediated calcium influx. Release of calcium from internal stores could trigger other processes that affect signaling from BGs to neurons. These two sources of calcium*

influx are not disambiguated in the manuscript. Also make clear which conclusions you are unable to draw from your data, rather than leave it open-ended.

Response: We thank the reviewer for this comment. We found D1R activation could induce Ca²⁺ from the internal store, or Ca²⁺ from Ca²⁺-permeable AMPARs. We believe that the rise of two types of Ca²⁺ signals may be temporally distinguishing, with the fast release from the internal store and the slow release from AMPARs. However, we do not have direct experimental evidence to address this assumption. We have mentioned this in the manuscript (page 19).

13. *The authors acknowledge previous studies that have shown the presence of slow depolarizing currents driven by activation of D2R on Purkinje cells, suggesting an autocrine function where dopamine that is released from PCs acts on the same cells. If dopamine is released from PCs and binds to surface expressed dopamine receptors in BGs, it could also bind to dopamine receptors on PCs. How does this affect interpretation of the author's results?*

Response: We are grateful that the reviewer brought up this comment. To determine if activation of D2R affects Ca²⁺ signals in BGs, we applied the D2R agonist quinpirole to GCaMP6f-expressing slices. Imaging of Ca²⁺ signals in BGs did not reveal any alteration (**Supplementary Fig. 5d**; also see the text in page 10). These additional data suggest that DA release from PCs and activation of D1Rs in BGs are not affected by D2R activation in PCs. Our data also showed that D2R expression is less than D1R in the cerebellum (**Supplementary Fig. 1a**), suggesting that the impact of D2R activation should be minimal. However, future experiments in our lab will determine if this low D2R expression in PCs has any effect on synaptic input to PCs and PC firing.

14. *The authors suggest that (based on cFos immunostaining in combination with a conditional knockout of D1R in BGs of the cerebellum not showing any impairments in behavior,) the cerebellum is not involved in delayed eyeblink conditioning. Even if this is what the authors find, it is a very contentious statement that is not backed up by most other studies. In fact, there is overwhelming empirical support for the involvement of the cerebellum in delayed eyeblink conditioning. The authors suggest that the interval used for conditioning may have deviated from other studies, but it is identical to the interval used in studies that clearly show cerebellar involvement. Moreover, several cells in the cerebellum are intrinsically active (Purkinje cells, cerebellar nuclear neurons to name a few). This will possibly impact the ability to make conclusive statements about neuronal activity in the cerebellum using cFos staining.*

Response: We apologize for the overstatement. To further address if the DA system is involved in the other form of eyeblink conditioning, in a new set of experiments we added trace eyeblink conditioning in which there is a stimulus-free interval between light stimulus and air-puff. Similar to the delay eyeblink conditioning, we did not see any difference in the percent CR and fraction eye closure between the controls and *Drd1a* cKO mice (**Fig. 6f and Supplementary Fig. 9b**). Because the protocols used in both delay and trace eyeblink conditioning are common, we intend to conclude that the DA system is simply not involved in eyeblink conditioning. We agree with the reviewer on this comment: PCs may be intrinsically active, so that c-fos staining is not

able to detect increased activity. We think it may also be true that eyeblink conditioning may not require strong PC activity, because in our hand, strong optogenetic stimulation can easily induce c-fos expression. The readers may further ask why D1R activation that can readily increase the number of PC spikes has no effect on eyeblink conditioning that only needs weak activation. We reason that the subcellular and molecular signaling required for eyeblink conditioning and other cerebellar functions, including social behavior, may be profoundly different. Further studies to address this issue will be extremely interesting. We have incorporated this discussion in the text (pages 21-22).

15. *The authors show that MLIs are activated after D1R activation in BGs, but the exact mechanism through which this happens remains unclear. This should be acknowledged.*

Response: Our result showing an increased inhibitory but a decreased excitatory input to PCs suggests an increase in glutamate release from BGs (see *Beppu, The Journal of Physiology 599, 2021* for glutamate release from BGs). However, we do not have direct evidence to prove it. We will explore this plausible mechanism in the future. We have acknowledged this point in the Discussion section (page 20).

16. *The authors state: “It should be noted that in a few cases especially in CF-PC synapses, there existed a transient increase in PSC amplitude lasting about 1-2 min.” We don’t see this data reported. It would be good to include the data as a supplemental figure.*

Response: We have plotted these data in a new graph (**Supplementary Fig. 7b**).

17. *The authors should more clearly emphasize throughout the text that the cKO is specific to BG in the cerebellar cortex (bearing in mind that astrocytes in other regions may potentially also be affected).*

Response: Thank the reviewer for pointing out this issue. We have highlighted this point in the text.

18. *In the transgenic animals use D1 receptors are selectively knocked out in BGs, but what do the authors believe is the likelihood that D1 receptor function in other astrocytes are also affected (e.g. <https://www.frontiersin.org/articles/10.3389/fnana.2017.00003/full>) and thus contribute to the reported impairments in learning and behavior? Some evidence of selectivity is presented by infusing D1R agonist directly into the cerebellum of WT mice (which may not have enough specificity to avoid also activating CN neurons expressing D1 receptors). However, this is only shown for the open field test.*

Response: We thank the reviewer for raising this important issue. We examined the expression of D1Rs in several other major brain regions (cerebellar cortex, hippocampus, and striatum) and found no significant expression of D1Rs in astrocytes. For more explanation on the immunochemical approach, we refer the reviewer to Point 7 in Reviewer 2. Based on these data, we believe that the major behavioral effect that *Drd1* cKO mice exhibit is caused by the deletion of D1Rs from the cerebellum. However, we cannot completely exclude the possibility of the effect on behavior of D1R deletion in astrocytes in other small brain regions, including SNc, as referenced by the reviewer. In

our future pharmacological experiments, as suggested by the reviewer, we should restrict our infusion to the smaller area by avoiding targeting to other brain areas including DCN. Following the treatment, we will then perform behavioral assessments, including social interaction. Nevertheless, we should bear in mind that D1R activation in a too small brain region may fail to induce a detectable behavioral change.

19. *Figure 5g shows a decline in firing frequency, but also change in the regularity of firing. Can the authors quantify firing regularity (e.g. CV2)?*

Response: We analyzed the CV2 of PC firing and found that it was increased after the SKF83822 treatment. These data suggest that the firing regularity is decreased following D1R activation. Loss of GABAergic inhibition on PCs has been known to increase their regularity, as reflected by decreased CV2 (see *Brown AM, Sci Rep 9, 2019*). This finding is consistent with ours as we found that D1R activation increases GABAergic activity and results in an increase in CV2.

20. *The authors mention achieving “near-super resolution” when studying localization of GluA1, GluA4, but do not provide figures for the exact resolution obtained (neither found in the main text, nor methods section). Please add this information.*

Response: LSM-800 Airyscan confocal microscope enabled us to obtain images with a near-super resolution. Since formatting and resizing images results in a loss of the resolution, we removed the statement of “near-super resolution” throughout the text.

21. *This manuscript really calls for a schematic to demonstrate the (putative) modes of action of locally released DA in the cerebellar circuit.*

Response: We have added a schematic diagram to the manuscript (**Fig. 8**). It summarizes the release of DA from PCs, the activation of D1Rs in BGs, the impact on excitatory (PF and CF) and inhibitory inputs to PCs, and the ultimate influence on PC firing.

Figures:

22. *In Figure 2 the box used to calculate background fluorescence can hardly be seen. Change color for clarity.*

Response: For clarity for all figures in the manuscript, we have changed all boxes where the dLight1.1 or Ca²⁺ signals were measured to the red boxes and all boxes where the background fluorescence was measured to the white or blue.

23. *Figure 3 panels (a) and (b) should show colors matching to those in other bar graphs.*

Response: We have changed the bar color so that these figures match all others (**Fig. 3a, c**).

24. *Supplemental Figure 5, please indicate which colors represent WT, or cKO.*

Response: We have added the boxes to show which is WT or *Drd1* cKO.

REVIEWER COMMENTS

Reviewer #1 (Remarks to the Author):

The authors have added new experiments that significantly improved the quality of the manuscript. In particular, the authors have convincingly demonstrated that chemogenetic activation of PCs induces increases in DA concentration in the molecular layer and that DA changes can occur spontaneously in vivo.

However, similarly to the first version of the MS the results shown in Fig 5 remain not convincing. The authors showed an inhibitory effect of SKF on the frequency of both spontaneous sEPSCs and sIPSCs with no alterations of their amplitudes. Moreover, they observed an inhibitory effect of D1R activation on PSC evoked by PF or CF activation. I remain convinced that the effect of D1R activation on IPSC and EPSC should be analysed on pharmacologically isolated evoked currents which is still not the case. Doing so, it would be proper to investigate the possible pre/post-synaptic origin of these SKF-induced effects by measuring a paired-pulse ratio.

In conclusion I suggest two options: either the required experiments are performed properly as suggested or another option I suggest moving these data to the Supplementary material and less emphasis on the possible conclusions.

Moreover, similarly to reviewer n°2, I'm surprised about the amplitude of CF-PSCs that are smaller than 500pA. This could suggest the use of very young mice for the electrophysiological study. I didn't find the information on mouse age in the M&M. I thus suggest the authors to discuss this point and include the information in the MS.

Minor:

I've added my minor points directly on the revised MS that is attached to this response.

Reviewer #2 (Remarks to the Author):

I read with interest the revised version of this manuscript, hoping that the authors would have heeded the referees' suggestions for improvements and additional experiments. While the manuscript is improved and the authors have clearly done a lot of new work, unfortunately, I remain unconvinced on several key points I raised in the first round of review. That said, I'm disinclined at this late stage, after so much work from the authors, to ask for more experiments.

1). The immunostaining in Figure 2 showing CYP2D in Purkinje cells, which is the cornerstone of the model, must be validated by a control with tissue from the existing CYP2D null mouse.

The authors were unable to procure this mouse (I made a single phone call and was able to get a commitment to liver them within 4 weeks). They argue for CYP2D in Purkinje cells with citations but these citations are unconvincing (blots rather than immunocytochemistry in some, immune-signals in Purkinje cells near background levels in others)

2). The staining of AAV-CAG-dlight1.1 in Figure 2 shows several cells types labelled, not just Bergmann glia. It then becomes problematic to attribute the dlight signal to Bergmann glia specifically.

Here, the authors have shown that dlight1.1 is not expressed in Purkinje cells but have not still shown that it is expressed in Bergmann glia specifically even though the experiments using dlight 1.1 measurements are used to support a model of DA action at Bergmann glia

3). In Figure 2DE, the dlight transient is evoked by extracellular electrode stimulation of the cerebellar cortex so it then becomes impossible to know if the dopamine originated from the Purkinje cells or some other cellular compartment. Intracellular Purkinje cell stimulation would be

illuminating here.

The authors were unable to evoke dlight transients with single Purkinje cell depolarization. Mass chemogenetic activation was required. To me this argues against a physiological role for Purkinje cell dopamine release.

4). Attributing the action of fluoxetine on dlight signals to CYP2D inhibition is risky when this is the only manipulation used targeting CYP2D and fluoxetine has other known pharmacological effects. The existing CYP2D null mouse would be useful here.

Again, the null mouse was not used. Instead a second nonspecific drug that inhibits CYP2D, quinidine, was employed. Two dirty drugs are not making a convincing case to me here.

5). If both SKF or chemogenetic activation of Bergmann glia are acting on Bergmann glial Ca transients downstream of depolarization then it should be possible to block these effects by hyperpolarizing the Bergmann glial cell with negative current injection.

Response to this point is fine.

6). In Figure 5D, the climbing fiber-evoked EPSCs seem to have an amplitude of about 300pA when $V_{hold} = -80$ mV (and there is no scale bar for Figure 5B). In my own experience, all-or-none CF-EPSCs in Cs-loaded Purkinje cells in this condition are at least 10-fold larger, to the point where they escape the voltage-clamp. I would need to see an I/O curve showing all-or-none responses to convince me that these are indeed CF-EPSCs.

Response to this point is fine.

7). The cre-lox strategy purported to selectively block the proposed cerebellar dopamine signaling pathway involves deleting the D1 receptor in cells expressing the glutamate transporter GLAST. While this will include Bergmann glial D1Rs, it will, of course, delete them in all cells of the body that express GLAST, including astrocytes throughout the central nervous system. It then becomes problematic to attribute the behavioral phenotypes specifically to actions at Bergmann glia.

Response to this point is fine.

8). I'm also left a bit confused about how the authors imagine this cerebellar dopamine signaling to operate in vivo. Dopamine release is presumably evoked by activating cerebellar Purkinje cells. Purkinje cells in vivo have a resting firing frequency of 40-100 Hz. Are we then to imagine that dopamine release from Purkinje cells is tonic, possibly modulated by firing frequency? Or do the authors see it as a phasic signal, perhaps only evoked by large Ca transients in the Purkinje cell as produced during a climbing fiber-evoked complex spike? The experiments to explore these key points are straightforward and use techniques that the authors have already mastered.

OK to let this point be settled by future experiments.

9). It's unsatisfying that the authors haven't tested the hypothesis that dopamine release from Purkinje cells is vesicular (as opposed to resulting from reverse transporter action). Applying vesicular monoamine transporter inhibitor drugs to Purkinje cells through the patch pipette while measuring dlight response to Purkinje cell activation would nail this point down.

Response to this point is fine.

Reviewer #3 (Remarks to the Author):

I appreciate the considerable extra effort the authors have invested in this paper to address the critiques. The points made in the initial review have, in my view, been adequately addressed.

We thank three reviewers for their positive evaluation of this revision. We appreciate their further comments, which make our manuscript more rigorous. We have given a point-by-point response to these comments as detailed below. The changes made to the manuscript are highlighted in yellow.

Reviewer #1

Major Points:

1. *However, similarly to the first version of the MS the results shown in Fig 5 remain not convincing. The authors showed an inhibitory effect of SKF on the frequency of both spontaneous sEPSCs and sIPSCs with no alterations of their amplitudes. Moreover, they observed an inhibitory effect of D1R activation on PSC evoked by PF or CF activation. I remain convinced that the effect of D1R activation on IPSC and EPSC should be analysed on pharmacologically isolated evoked currents which is still not the case. Doing so, it would be proper to investigate the possible pre/post-synaptic origin of these SKF-induced effects by measuring a paired-pulse ratio. In conclusion I suggest two options: either the required experiments are performed properly as suggested or another option I suggest moving these data to the Supplementary material and less emphasis on the possible conclusions.*

Response: The alteration of PPR of EPSCs or IPSCs is a good indication of presynaptic modification, in addition to several other electrophysiological (e.g., evaluation of synaptic fatigue and of the ratio of action potential-evoked to hypertonic sucrose-induced charge transfer) and optical (e.g. pH-sensitive optical probes and quantum dots) approaches. Since the determination of presynaptic mechanism is not the primary aim in our study, to make this manuscript more succinct we abridged the description of how we reached to our finding (pages 12-13), and as suggested by the reviewer, moved several corresponding panels to the supplementary material. We appreciate the alternative that the reviewer offered to us. We hope that such change has made this manuscript more concise and understandable.

2. *Moreover, similarly to reviewer n°2, I'm surprised about the amplitude of CF-PSCs that are smaller than 500pA. This could suggest the use of very young mice for the electrophysiological study. I didn't find the information on mouse age in the M&M. I thus suggest the authors to discuss this point and include the information in the MS.*

Response: To address the issue of the small amplitude of CF-evoked PSCs, we first examined if the electrode is correctly placed. To do this, the anterograde tracer Dextran Alexa Fluor-594 was injected into the inferior olive for labeling CF afferents in the cerebellum. Mice were used for experiments 4 d later. We found that CF afferents nearly cross the entire GCL (**Supplementary Fig. 7c**). Consistently, we generally placed our electrodes in the middle of the GCL. Following the verification of electrode placement, we then determined if the size of PSCs is related to the age of animals. In our studies, we used animals at age of 2-3 months (PND60-90). Here, we compared young (~PND30) and old (~PND90) animals (**Supplementary Fig. 7d**). We found that there was no difference between these two groups. With the use of our Cs-based intracellular solution, we could induce nearly all-or-none PSCs (see 20 and 30 μ A, **Supplementary Fig. 7d**). However, the range of stimulus intensity is rather narrow. A slight increase of stimulus intensity (40 μ A) could result in unclamped PSCs. We also found that in our

hands, before PSCs overshoot, the PSC amplitude is generally less than 500 pA. We tested if the small size of PSCs is due to the composition of the intracellular solution. We added the Na⁺ channel blocker QX314 to the intracellular solution. We showed that large PSCs (> 500pA) can be reliably induced (**Supplementary Fig. 7d**). Based on these observations, we are confident that these small PSCs are indeed evoked by stimulation of CFs and that the size of PSCs is related to the composition of the intracellular solution

Minor Points:

3. *It seems that there is YFP labelling in the cerebellum in Supplementary Figure 3C.*

Response: We thank the reviewer for the thorough examination of our immunostaining images. Sparse YFP labelling in the deep cerebellar nuclei is likely due to the infection of some DAT-expressing neurons by off-target AAV-DIO-YFP. Our original observation shown in the manuscript demonstrates that DA in the cerebellar cortex is less likely to be supplied by neurons in the LC, considering the lack of axonal projections into the cerebellum. These potential isolated DA neurons in the deep cerebellar nuclei also do not appear to send projections to the cerebellar cortex. Collectively, in addition to the LC, the potential DA neurons in the deep cerebellar nuclei also do not provide DA to D1R-expressing BGs. To avoid creating the confusion, we did not include this data in the text, but we modified the sentence to emphasize the lack of axonal projections from the LC to the cerebellum (page 6), which is the major point that the readers are most likely to ask to be clarified.

4. *“Interestingly, the hyperpolarization itself also increased Ca²⁺ activity (Supplementary Fig. 5a)”. How is this possible? This is counter-intuitive and I_h is unlikely expressed in BG.*

Response: I_h channels are generally known to be expressed in neurons. However, a few studies also showed that I_h channels can be expressed in cortical astrocytes (*Guatteo E. Glia 16, 1996; Honsa P. Glia 62, 2014*). It will be interesting to determine whether they are also expressed in BGs. As many extracellular and intracellular molecules can trigger Ca²⁺ rise, whether they can be activated by during BG membrane hyperpolarization is also an interesting topic for our future investigation. Here, to show that the possibility of the expression of I_h channels, we cited one more paper (page 9).

5. *It's not really a “block” of spontaneous Ca²⁺ transients ... a reduction of the frequency and no significant effect on the amplitude of deltaF/F.*

Response: We appreciate the reviewer for pointing this out. We have reworded this sentence to reflect our statistical analysis more accurately (page 10).

6. *Instead: high Cl⁻ solution; Instead: low Cl⁻ solution.*

Response: We have revised the manuscript appropriately (page 13).

7. *Are presynaptic glutamate receptors ionotropic or metabotropic? This sentence is too faded. Which kind of glutamate receptors expressed on PF and CF may explain the effects on EPSCs*

Response: The results showing increased sEPSCs frequency but unaltered sEPSC amplitude suggest presynaptic modification. Previous studies demonstrated that activation of presynaptic NMDARs (*Lonchamp E. PLoS One 7, 2012*) and KARs (*Delaney AJ. Neuron 36, 2002*) can rapidly alter glutamate release to PCs. In light of these findings, we speculate that presynaptic ionotropic GluRs may be activated during the early action of D1Rs. More experiments in the future are needed to confirm this hypothesis. Regarding the prolonged effect on PF and CF presynaptic terminals, we suggest that mGluRs may be activated. The autoreceptors Group II and III mGluRs at presynaptic sites are well known to decrease neurotransmitter release (*Pinheiro PS. Nat Rev Neurosci 9, 2008*). Consistently, the Group III mGluRs and their effect on glutamate release have been revealed in PF-PC synapses (*Lorez M. Br J Pharmacol 138, 2002*). We have rephrased the sentence and added the references.

8. What is the PND of the mice?

Response: We are sorry for missing this information in the text. We used both male and female animals at age of 2-3 months. We have incorporated it into the text.

9. The topographical error in the Reference.

Response: Corrected (page 53).

10. Orange arrow = stimulating electrode position.

Response: We have given the annotation to all symbols in this panel (pages 59-60)

11. To replace “treated with SKF83822” with “before and during SKF83822 application”.

Response: We thank the reviewer for this point. We have changed it (page 65).

12. Glutamate release by BG is a pure hypothesis here...; add “may”; this has not been proven.

Response: Indeed, this is speculation. Although we discussed this possibility in the text (page 20), we do not have direct evidence to prove it. We have changed our description accordingly.

Reviewer #2

1. *The authors were unable to procure this mouse (I made a single phone call and was able to get a commitment to liver them within 4 weeks). They argue for CYP2D in Purkinje cells with citations but these citations are unconvincing (blots rather than immunocytochemistry in some, immune-signals in Purkinje cells near background levels in others)*

Response: We are thankful for the reviewer's comments on our revised manuscript. With the further extensive investigation by our group, we found two ideal null mouse strains from Taconic (<https://www.taconic.com/mouse-model/cyp2d-knockout-mouse>) and Jackson Laboratories (<https://www.jax.org/strain/002910>). These mice are deficient in *Cyp2d* genes, including the mouse ortholog of *Cyp2d6*, *Cyp2d22*, relevant in the metabolism of tyramine to dopamine in the brain. We have contacted both vendors by phone and found that the only method of obtaining such mice is cryorecovery. This

process will take an extensive time (12-16 wks). It also takes an equal time duration to breed these animals. We completely agree that using the *Cyp2d* KO mice will provide confirmative evidence for the expression of CYP2Ds in PCs and its role in dopamine release. However, due to the time restraint, we sincerely apologize for not being able to collect data from these animals. We have explicitly pointed out this weakness in the text (page 19).

Although lacking the use of *Cyp2d* null mice, we performed additional experiments to support our conclusion. In our first immunostaining experiments, we used an antibody against N-terminal CYP2D6 (RPPVPITQILGFGPRSQGVFLARYGPAWREQRRFSVSTLRNLGLGKKSLE; <https://www.sigmaaldrich.com/US/en/product/sigma/av41675>). In our revision, we added another antibody against C-terminal CYP2D6 (AWGLLLMILHPDVQRRVQQEIDDVIGQVRRPEM; <https://www.thermofisher.com/antibody/product/CYP2D6-Antibody-Polyclonal/PA5-79129>). The second set of experiments also showed that CYP2Ds are expressed in PCs (**Supplementary Fig. 4a**), confirming our previous results (**Fig. 2a**).

2. *Here, the authors have shown that dLight1.1 is not expressed in Purkinje cells but have not still shown that it is expressed in Bergmann glia specifically even though the experiments using dLight 1.1 measurements are used to support a model of DA action at Bergmann glia.*

Response: We have demonstrated that dLight1.1 is largely expressed in BGs, but not in PCs (**Supplementary Fig. 4b**). Consistent with this result, both DA application and electric stimulation in the cerebellum could induce large dLight1.1 signals (**Fig. 2c, d**). We believe that dLight1.1 signals are primarily detected from BGs, which supports our model of DA action on BGs. However, as the dLight1.1 viral vector is not driven by the BG-specific promoter, we cannot completely exclude the possibility of a small contribution from other cell types. In our future experiments, we will perform similar experiments to confirm this model when the virus that expresses dLight1.1 specifically in BGs is commercially available. We apologize for not having the capability to address this issue directly. We have mentioned this weakness in this revision (page 19).

3. *The authors were unable to evoked dlight transients with single Purkinje cell depolarization. Mass chemogenetic activation was required. To me this argues against a physiological role for Purkinje cell dopamine release.*

Response: We thank the reviewer for raising this issue. After an extensive investigation, we found that there were not many studies measuring DA release at a single-cell level. These studies generally used fast cyclic voltammetry (FSCV) in cultured cells. Compared with those measured in slices or *in vivo* animals, DA signals generally are very small (e.g., Kozminski K, *Anal Chem* 70, 1998). Considering that the spontaneous dLight1.1 signals in the cerebellum are small (**Fig. 2i**), we are inclined to state that action potentials from a single PC are insufficient to induce detectable DA signals. In the cerebellum of freely moving mice, PF activity-related simple spikes in PCs occur at 40-100Hz, and CF activity-related complex spikes occur at 1Hz. Also, many PCs typically fire synchronously. With such intense activity from PCs, DA release is likely to occur. Our data showed that the chemogenetic stimulation induces action potentials in

PCs at a level similar to spontaneous *in vivo* PC activity (**Fig. 2f**). Therefore, we believe that chemogenetically-induced activity is equivalent to physiological activity, which can explain the role of DA in our model. In addition to chemogenetics, we also attempted to apply optogenetic stimulation because it allows for the timing control of PC firing. We expressed ChrimsonR in PCs in combination with dLight1.1. We prepared acute slices from these animals (n = 32 slices/8 mice) and introduced orange light (595 nm) to activate ChrimsonR during detecting dLight1.1 signals by blue light (470 nm). However, one light often interferes with the other, confounding the dLight1.1 signals. Therefore, we did not add these data to this revision. We believe chemogenetics is an ideal tool to detect DA signals released specifically from PCs.

4. *Again, the null mouse was not used. Instead a second nonspecific drug that inhibits CYP2D, quinidine, was employed. Two dirty drugs are non making a convincing case to me here.*

Response: We thank the reviewer for raising this issue. Currently, there are no highly selective blockers for CYP2D. Indeed, testing our hypothesis in null mice is optimal. Unfortunately, as pointed out in Question 1, presently, there are no *Cyp2d* null mice readily available. Although we provided additional evidence, as described above, we sincerely apologize for not being able to resolve this issue specifically. We have discussed this weakness in the revised manuscript (page 19).

Reviewer #3

We thank the reviewer for the approval of our revised manuscript.